# Sim and Real: Better Together

**Shirli Di Castro Shashua** *
Technion Institute of Technology
Haifa, Israel
shirlidi@technion.ac.il

**Shie Mannor**
Technion and NVIDIA Research
Israel
shie@technion.ac.il
smannor@nvidia.com

**Dotan Di Castro**
Bosch Center of AI
Haifa, Israel
dotan.dicastro@il.bosch.com

## Abstract

Simulation is used extensively in autonomous systems, particularly in robotic manipulation. By far, the most common approach is to train a controller in simulation, and then use it as an initial starting point for the real system. We demonstrate how to learn simultaneously from both simulation and interaction with the real environment. We propose an algorithm for balancing the large number of samples from the high throughput but less accurate simulation and the low-throughput, high-fidelity and costly samples from the real environment. We achieve that by maintaining a replay buffer for each environment the agent interacts with. We analyze such multi-environment interaction theoretically, and provide convergence properties, through a novel theoretical replay buffer analysis. We demonstrate the efficacy of our method on a sim-to-real environment.

## 1 Introduction

Reinforcement learning (RL) is a framework where an agent interacts with an unknown environment, receives a feedback from it, and optimizes its performance accordingly [44, 3]. There have been attempts of learning a control policy directly from real world samples [28, 49, 36, 21]. However, in many cases, learning from the actual environment may be slow, costly, or dangerous, while learning from a simulated system can be fast, cheap, and safe. The advantages of learning from simulation are counterbalanced by the *reality-gap* [18]: the loss of fidelity due to modeling limitations, parameter errors, and lack of variety in physical properties. The quality of the simulation may vary: when the simulation mimics the reality well, we can train the agent on the simulation and then transfer the policy to the real environment, in a one shot manner (e.g., [2]). However in many cases, simulation demonstrates low fidelity which leads to the following question: *Can we mitigate the differences between real environments ("real") and simulations ("sim") thereof, so as to train an agent that learns from both, and performs well in the real one?*

In this work, we propose to learn simultaneously on real and sim, while controlling the rate in which we collect samples from each environment and controlling the rate in which we use these samples in the policy optimization. This synergy offers a speed-fidelity trade-off and harnesses the advantage of each domain. Moreover, the simulation speed encourages exploration that helps to accelerate the learning process. The real system in turn can improve exploitation in the sense that it mitigates the challenges of sim-to-real policy transfer, and encourages the learner to converge to relevant solutions.

---

*This research was conducted during an internship in Bosch Center of AI.

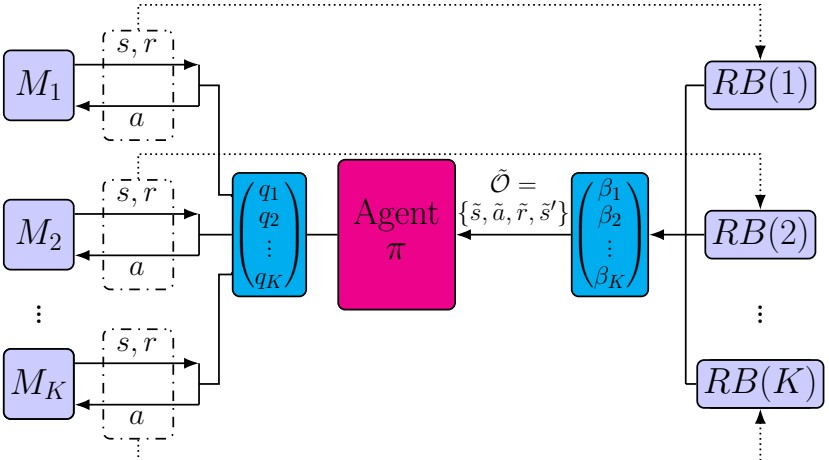

Figure 1: Mixing $K$ environments scheme. The agent selects an environment $M_i$ with probability $q_i$ and interacts with it. Simultaneously, the agent chooses RB(j) with probability $\beta_j$ and samples from this replay buffer a stored transition $\tilde{\mathcal{O}}$, which is used for estimating the TD error and update the policy parameters.

A general scheme describing our proposed setup is depicted in Figure 1. In a nutshell, there is a single agent interacting with $K$ environments (on the left). Each sample provided by an environment is pushed into a corresponding replay buffer (RB). On the right, the agent pulls samples from the RBs and is trained on them. In the sim-to-real scheme, $K = 2$.

In the specific scheme for mixing real and sim samples in the learning process, separate probability measures for collecting samples and for optimizing parameters policies are used. The off-policy nature of our scheme enables separation between real and sim samples which in turn helps controlling the rate of real samples used in the optimization process. In this work we discuss two RL algorithms that can be used with this scheme: (1) off-policy linear actor critic with mixing sim and real samples and (2) Deep Deterministic Policy Gradient (DDPG; [29]) mixing scheme variant based on neural networks. We analyze the asymptotic convergence of the linear algorithm and demonstrate the mixing samples variant of DDPG in a sim-to-real environment.

The naive approach in which one pushes the state-action-reward-next-state tuples into a single shared replay buffer is prone to failures due to the imbalance between simulation and real roll-outs. To overcome this, we maintain separate replay buffers for each of the environments (e.g., in the case of a single robot and a simulator we would have two replay buffers). This allows us to extract the maximum valuable information from reality by distinguishing its tuples from those generated by other environments, while continuously improving the agent using data from all input streams. Importantly, although the rate of samples is skewed in favor of the simulation, the learning may be carried out using a different rate. In a sense, the mechanism we suggest is a version of the *importance sampling* technique [10].

Our main contributions in this work are as follows:

1. We present a method for incorporating real system samples and simulation samples in a policy optimization process while distinguishing between the rate of collecting samples and the rate of using them.

2. We analyze the asymptotic convergence of our proposed mixing real and sim scheme.

3. To the best of our knowledge, we provide for the first time theoretical analysis of the dynamics and properties of replay buffer such as its Markovity and the explicit probability measure induces by the replay buffer.

4. We demonstrate our findings in a simulation of sim-to-real, with two simulations where one is a distorted version of the other and analyze it empirically.

## 2 Related Work

**Sim-to-Real:** Sim-to-Real is a long investigated topic in robotics where one aims to reduce the reality gap between the real system and its digital twin implementation. A general framework where we transfer results from one domain to another is *domain adaptation*. In vision, this approach have helped to gain state-of-the-art results [15, 43, 32, 9, 23, 42]. In our work, we focus on the physical aspects of the sim-to-real gap. Related to domain adaptation, is the approach of *domain randomization*, where the randomization is done in simulation in order to robustify and enhance the detection and object recognition capability [45, 40, 19, 47]. Recently, James et al. [20] proposed a method where both simulation and reality are adapted to a common domain. Andrychowicz et al. [2] extensively randomize the task of reaching a cube pose where one-shot transfer is achieved but with large sample complexity. Randomization may also be applied to dynamics, e.g., [35], where robustness to inaccuracy in real world parameters is achieved.

Another approach in Sim-to-Real is how to change the simulation in the light of real samples. In [11] the agent learns mainly from simulation but its parameters are updated to match the behavior in reality by reducing the difference between simulation and reality roll-outs. Our method is a direct approach that incorporates phenomena that is difficult to simulate accurately. In Bayesian context, [39] provide a principled framework to reason about the uncertainty in simulation parameters. Kang et al. [22] investigated how real system and simulation data can be combined in training deep RL algorithms. They separate between the data types by using real data to learn about the dynamics of the system, and simulated data to learn a generalizing perception system. Our method mix real and simulation data by controlling the rate of streaming each data type into the learning agent.

**Replay Buffer analysis:** Large portion of RL algorithms use replay buffers [30, 33] but here we review only works that provide some analysis. Several works study the effect of replay buffer size on the agent performance [51, 31]. Our focus is the effect of controlling the rate of collecting samples and the rate of using them in the optimization process. Fedus et al. [14] investigated the effect of the ratio between these rates on the learning process through simulated experiments, while our focus is on the theoretical aspects. Other works studied the criteria for prioritizing transitions to enhance learning [41, 34, 50]. In case of multiple agents that share their policy, Horgan et al. [16] argue in favor of a shared replay buffer for all agents and a prioritizing mechanism. We, on the other hand, emphasize the advantage of separating replay buffers when collecting samples from different environments to enable a mixing management in the learning process.

**Stochastic Approximation:** Our proposed algorithm is based on the Stochastic Approximation method [27]. Konda and Tsitsiklis [25] proposed the actor-critic algorithm, and established the asymptotic convergence for the two time-scale actor-critic, with TD($\lambda$) learning-based critic. Bhatnagar et al. [6] proved the convergence result for the original actor-critic and natural actor-critic methods. Di Castro and Meir [13] proposed a single time-scale actor-critic algorithm and proved its convergence. Recently, several finite sample analyses were applied by [48, 52, 12] and more but these works have not analyzed the replay buffer asymptotic behavior while we do.

## 3 Setup

We model the problem using a Markov Decision Process (MDP; [38]), where $\mathcal{S}$ and $\mathcal{A}$ are the state space and action space, respectively. We let $P(s'|s, a)$ denote the probability of transitioning from state $s \in \mathcal{S}$ to state $s' \in \mathcal{S}$ when applying action $a \in \mathcal{A}$. The MDP measure $P(s'|s, a)$ and the policy measure $\pi_\theta(a|s)$ induce together a Markov Chain (MC) measure $P_\theta(s'|s)$ ($P_\theta$ is matrix form). We consider a probabilistic policy $\pi_\theta(a|s)$, parameterized by $\theta \in \Theta \subset \mathbb{R}^d$ which expresses the probability of the agent to choose an action $a$ given that it is in state $s$. We let $\mu_\theta$ denote the stationary distribution induced by the policy $\pi_\theta$. The reward function is denoted by $r(s, a)$. Throughout the paper we assume the following.

**Assumption 1.** *1. The set $\Theta$ is compact. 2. The reward $|r(\cdot, \cdot)| \leq 1$ for all $s \in S, a \in A$.*

**Assumption 2.** *For any policy $\pi_\theta$, the induced Markov chain of the MDP process $\{s_\tau\}_{\tau \geq 0}$ is irreducible and aperiodic.*

The goal of the agent is to find a policy that maximizes the *average reward* that the agent receives during its interaction with the environment [38]. Under an ergodicity assumption, the average reward

over time eventually converges to the expected reward under the stationary distribution [3]:

$$\eta_\theta \triangleq \lim_{T \to \infty} \frac{\sum_{\tau=0}^{T} r(s_\tau, a_\tau)}{T} = \mathbb{E}_{s \sim \mu_\theta, a \sim \pi_\theta}[r(s, a)]. \tag{1}$$

The state-value function evaluates the overall expected accumulated rewards given a starting state $s$ and a policy $\pi_\theta$

$$V^{\pi_\theta}(s) \triangleq \mathbb{E}\left[\sum_{\tau=0}^{\infty} (r(s_\tau, a_\tau) - \eta_\theta) \middle| s_0 = s, \pi_\theta\right], \tag{2}$$

where the actions follow the policy $a_\tau \sim \pi_\theta(\cdot|s_\tau)$ and the next state follows the transition probability $s_{\tau+1} \sim P(\cdot|s_\tau, a_\tau)$. Denote $\mathbf{v}^\theta$ to be the vector value function defined in (2). Therefore, the vectorial Bellman Equation (BE) for a fixed policy $\pi_\theta(\cdot|\cdot)$ is $\mathbf{v}^\theta = r - \eta_\theta + P_\theta \mathbf{v}^\theta$, where $r$ is a vector of rewards for each state [38]. We recall that the solution to the BE is unique up to an additive constant. In order to have a unique solution, we choose a state $s^*$ to be of value 0, i.e., $V^{\pi_\theta}(s^*) = 0$ (due to Assumption 2, $s^*$ can be any of $s \in S$).

In our specific setup, we consider a model where there are $K$ MDPs, denoted by $M_k$, all share the same state space $\mathcal{S}$, action space $\mathcal{A}$, and reward function $r(s, a)$. The environment dynamics, though, are different, and are denoted by a transition function $P_k(\cdot|\cdot, \cdot)$. Together with a shared policy $\pi_\theta(\cdot|\cdot)$, each $M_k$ is induced by a state transition measure $P_{\theta,k}(s'|s)$ and a stationary distribution $\mu_{\theta,k}$. Let $\eta_{\theta,k} = \mathbb{E}_{s \sim \mu_{\theta,k}, a \sim \pi_\theta}[r(s, a)]$ and define the average reward over $K$ environments,

$$\bar{\eta}_\theta = \mathbb{E}_{k \sim \beta, s \sim \mu_{\theta,k}, a \sim \pi_\theta}[r(s, a)] = \sum_{k=1}^{K} \beta_k \eta_{\theta,k}, \tag{3}$$

where $\beta$ is a distribution which will be defined in Section 4. The following assumption resembles Assumption 2 for $K$ environments.

**Assumption 3.** *For any policy $\pi_\theta$, the induced Markov chain of MDP $M_k$ is irreducible and aperiodic for all $k = 1 \ldots K$.*

We define $\nu_k$ to be the *throughput* of $M_k$ and it is defined as the number of samples MDP $M_k$ provides for a unit time. In sim-to-real context, this setup can practically handle several robots and several simulation instances. We assume for the sim-to-real scenario that $\nu_s > \nu_r$.

Since the samples from real arrive at a lower throughput than the sim, if we push the samples into two separate Replay Buffers (RB; [30, 33]) based on their sources, we can leverage the relatively scarce, but valuable samples that originated in the real system. This observation is the main motivation for our "Mixing Sim and Real" scheme, presented in the next section.

## 4 Mixing Sim and Real Algorithm

In order to reconcile the dynamics disparity, we propose our *Mixing Sim and Real Algorithm with Linear Actor Critic*, presented in Algorithm 1 and described in Figure 1. We consider $K$ environments, modeled as MDPs, $M_1, \ldots, M_K$, where the agent maintain a replay buffer $RB(k)$ for each MDP, respectively. For the sake of analysis simplicity, we replace $\{\nu_k\}$ with the following random variable. The agent chooses an environment to communicate with according to $I \sim Categorial(q_1, \ldots, q_K)$ where $q \triangleq [q_1, \ldots, q_K]$, $q_i \geq 0$, and $\sum_i q_i = 1$. The agent collects transitions $\{s_i, a_i, r_i, s'_i\}$ from the chosen environment and stores them in the corresponding $RB(i)$. In order to approximate the rates $\{\nu_k\}_{k=1}^{K}$ correctly, we choose $q_i = \nu_i / \sum_k \nu_k$ for the agent to interact according to the rates.

We train the agent in an off-policy manner. The agent selects $RB(j)$ for sampling the next batch for training according to $J \sim Categorial(\beta_1, \ldots, \beta_K)$ where $\beta \triangleq [\beta_1, \ldots, \beta_K]$, $\beta_j \geq 0$, and $\sum_j \beta_j = 1$. This distribution remains static, and hence the selections in time are i.i.d[2]. In addition, the $\beta$ distribution that selects which samples to train over should be different than the $q$ distribution that controls the throughput each environments pushes samples to the RB. In that way, scarce samples from the real environment can get higher influence on the training.

---

[2]We note that one could remove this restriction and think of other schemes in which the replay buffer selection distribution changes over time based on some prescribed optimization goal, cost, etc.

---

**Algorithm 1** Mixing Sim and Real with Linear Actor Critic

---

1: Initialize Replay Buffers $RB(k)$ with size $N$ and initialize $t_k = 0$ for $k = 1, \ldots K$.
2: Initialize actor parameters $\theta_0$, critic parameters $v_0$ and average reward estimator $\eta_0$.
3: **for** $\tau = 0, \ldots$ **do**
4:     Sample $i \sim q$, interact with $M_i$ according to policy $\pi_{\theta_\tau}$ and add the transition $\{s_{i,t_i}, a_{i,t_i}, r_{i,t_i}, s_{i,t_i+1}\}$ to $RB(i)$. Increment $t_i \leftarrow t_i + 1$.
5:     Sample $j \sim \beta$ and choose $N_{\text{batch}}$ transitions from $RB(j)$ denoted as $\{\tilde{\mathcal{O}}_{j,n}^z(\tau)\}_{z=1}^{N_{\text{batch}}}$.
6:     $\delta(\tilde{\mathcal{O}}^z) = \tilde{r}^z - \eta_\tau + \phi(\tilde{s}'^z)^\top v_\tau - \phi(\tilde{s}^z)^\top v_\tau$
7:     Update average reward
    $\eta_{\tau+1} = \eta_\tau + \alpha_\tau^\eta(\frac{1}{N_{\text{batch}}}\sum_z \tilde{r}^z - \eta_\tau)$
8:     Update critic $v_{\tau+1} = v_\tau + \alpha_\tau^v \frac{1}{N_{\text{batch}}}\sum_z \delta(\tilde{\mathcal{O}}^z)\phi(\tilde{s}^z)$
9:     Update actor $\theta_{\tau+1} = \Gamma\left(\theta_\tau - \alpha_\tau^\theta \frac{1}{N_{\text{batch}}}\sum_z \delta(\tilde{\mathcal{O}}^z)\nabla_\theta \log \pi_\theta(\tilde{a}^z|\tilde{s}^z)\right)$
10: **end for**

---

Once a RB is selected, the sampled batch is used for optimizing the actor and the critic parameters. In this work, we propose a two time scale linear actor critic optimization scheme [25], which is an RB-based version of [6] Algorithm. We analyze its convergence properties in Section 5. We note, however, that other optimization schemes can be provided, such as DDPG [29], which we use in our experiments.

We define a tuple of indices $(k, n)$ where $k$ corresponds to $RB(k)$ and $n$ corresponds to the $n$-th sample in this $RB(k)$. In addition, it corresponds to time $t(k, n)$ where this is the time when the agent interacted with the $k$-th MDP and the $n$-sample was added to $RB(k)$. Let $\tilde{\mathcal{O}}_{k,n}(\tau) \triangleq \{\tilde{s}_{k,n}, \tilde{a}_{k,n}, \tilde{r}_{k,n}, \tilde{s}'_{k,n}\}$ be a transition sampled at time $\tau$ from $RB(k)$. Whenever it is clear from the context, we simply use $\tilde{\mathcal{O}}$.

The temporal difference (TD) error $\delta(\tilde{\mathcal{O}})$ is a random quantity based on a single sampled transition from $RB(k)$, $\delta(\tilde{\mathcal{O}}) = r(\tilde{s}, \tilde{a}) - \eta + \phi(\tilde{s}')^\top v - \phi(\tilde{s})^\top v$, where $\hat{V}_v^{\pi_\theta}(s) = \phi(s)^\top v$ is a linear approximation for $V^{\pi_\theta}(s)$, $\phi(s) \in \mathbb{R}^d$ is a feature vector for state $s$ and $v \in \mathbb{R}^d$ is a parameter vector. In Algorithm 1, average reward, critic and actor parameters are updated based on the TD error (see lines 7 - 9). Note that for the actor updates, we use a projection $\Gamma(\cdot)$ that projects any $\theta \in \mathbb{R}^d$ to a compact set $\Theta$ whenever $\theta \notin \Theta$.

In order to gain understanding of our proposed setup, in the next section we characterize the behaviour of the iterations in Algorithm 1.

## 5  Convergence Analysis for Mixing Sim and Real with Linear Approximation

The standard tool in the literature for analyzing iterations of processes such as two time scale Actor-Critic in the context of RL is *SA; Stochastic Approximation* [26, 7, 4]. This analysis technique includes two parts: proving the existence of a fixed point, and bounding the rate of convergence to this fixed point. By far, the most popular methods for proving convergence is the *Ordinary Differential Equation (ODE) method*. Usually, the iteration should demonstrate either some monotonicity property, or a contraction feature in order for the iteration to converge.

Although in practice such algorithms (after some tuning) usually converge to an objective value, it is not always guaranteed. To achieve that in a stochastic approximation setup, the main known result shows that the iteration can be decomposed into a deterministic function, which depends only on the problem parameters, and a martingale difference noise, which is bounded in some way.

In this section we show that the iterations of Algorithm 1 converge to a stable point of a corresponding ODE. We begin with showing that the process of sampling transitions from RBs is a Markov process. Afterward, we show that if the original Markov chain is irreducible and aperiodic, then also the RBs Markov process is irreducible and aperiodic. This property is required for proving the convergence of the iterations in Algorithm 1 using SA tools. We conclude this section with showing that if in some

sense sim is close to real, then the properties of the mixed process are close to the properties of both sim and real.

## 5.1 Asymptotic Convergence of Algorithm 1

Let $RB(k)$ be a replay buffer storing the last $N$ transitions from MDP $k$. Let $RB_\tau(k)$ be the state of $RB(k)$ at time $\tau$, i.e., $RB_\tau(k) \triangleq \{\mathcal{O}_{k,1}, \ldots, \mathcal{O}_{k,N}\}$, where $\mathcal{O}_{k,n} = \{s_{k,n}, a_{k,n}, r_{k,n}, s'_{k,n}\}$ is a transition tuple pushed at some time $t(k,n) < \tau$. We denote the collection of all $RB_\tau(k)$ as $\bigcup_{k=1}^{K} RB_\tau(k)$. We define $I_\tau$ and $J_\tau$ be i.i.d random processes based on $I$ and $J$, respectively. We define $Y_\tau$ to be the process induced by Algorithm 1, i.e.,

$$Y_\tau = \left[ \bigcup_{k=1}^{K} RB_\tau(k), I_\tau, J_\tau \right]. \tag{4}$$

The next lemma states the $Y_\tau$ is Markovian. The proof is deferred to the Supplementary material A.1.

**Lemma 1** ($Y_\tau$ induced by Algorithm 1 is Markovian)**.** *1. The random process $Y_\tau$ is a Markovian. 2. Under Assumption 3, there exists some $\tau' > 0$ such that $Y_\tau$ is irreducible and aperiodic for $\tau \geq \tau'$.*

Next, we present several assumptions that are necessary for proving the convergence of Algorithm 1. The first assumption is a standard requirement for policy gradient methods.

**Assumption 4.** *For any state–action pair $(s,a)$, $\pi_\theta(a|s)$ is continuously differentiable in the parameter $\theta$.*

Proving convergence for a general function approximation is hard. In our case we demonstrate the convergence for a linear function approximation (LFA; [4]). In matrix form, it can be expressed as $V = \Phi v$ where $\Phi \in \mathbb{R}^{|S| \times d}$. The following assumption is needed for the uniqueness of the convergence point of the critic.

**Assumption 5.** *1. The matrix $\Phi$ has full rank. 2. The functions $\phi(s)$ are Liphschitz in $s$ and bounded. 3. For every $v \in \mathbb{R}^d$, $\Phi v \neq e$ where $e$ is a vector of ones.*

In order to get a *with probability 1* using the SA convergence, the following standard assumption is needed. Note that in the actor-critic setup we need two time-scales convergence, thus, in this assumption the critic is a 'faster' recursion than the actor.

**Assumption 6.** *The step-sizes $\{\alpha_\tau^\eta\}$, $\{\alpha_\tau^v\}$, $\{\alpha_\tau^\theta\}$, $\tau \geq 0$ satisfy $\sum_\tau^\infty \alpha_\tau^\eta = \sum_\tau^\infty \alpha_\tau^v = \sum_\tau^\infty \alpha_\tau^\theta = \infty$, $\sum_\tau^\infty (\alpha_\tau^\eta)^2$, $\sum_\tau^\infty (\alpha_\tau^v)^2$, $\sum_\tau^\infty (\alpha_\tau^\theta)^2 < \infty$ and $\alpha_\tau^\theta = o(\alpha_\tau^v)$.*

We define the induced MC for the time $t(k,n)$ with a corresponding parameter $\theta_{t(k,n)}$. For this parameter, we denote with $P_{t(k,n)}$ the transition matrix at that time and the corresponding state distribution vector $\rho_{t(k,n)}$ (both induced by the policy $\pi_{\theta_{t(k,n)}}$). Finally, we define the following diagonal matrix $S_{t(k,n)} \triangleq \mathrm{diag}(\rho_{t(k,n)})$ and the reward vector $r_{t(k,n)}$ with elements $r_{t(k,n)}(s) = \sum_a \pi_{\theta_{t(k,n)}}(a|s)r(s,a)$. Based on these definitions we define

$$A_\tau \triangleq \sum_{k=1}^{K} \sum_{n=1}^{N} \frac{\beta_k}{N} S_{t(k,n)} \left( P_{t(k,n)} - \mathbf{I} \right), \quad b_\tau \triangleq \sum_{k=1}^{K} \sum_{n=1}^{N} \frac{\beta_k}{N} S_{t(k,n)} \left( r_{t(k,n)} - \eta_{\theta,k} e \right). \tag{5}$$

where $\mathbf{I}$ is the identity matrix and $e$ is a vector of ones. The intuition behind $A_\tau$ and $b_\tau$ is the following. For an online TD(0)-learning under a stationary policy we have a fixed point at the solution to the equation $\Phi^\top D(P - \mathbf{I})\Phi v + \Phi^\top D(r - \eta) = 0$ ([4]; Lemma 6.5). In our case, since we have $K$ RBs where each one with $N$ samples entered at different times, we have a superposition of all these samples. When $\tau \to \infty$, $\rho_{t(k,n)} \to \mu_{\theta,k}$ for all index $n$. We let $S_{\theta,k} \triangleq \mathrm{diag}(\mu_{\theta,k})$ and define

$$A_\theta \triangleq \sum_{k=1}^{K} \beta_k S_{\theta,k} \left( P_{\theta,k} - \mathbf{I} \right), \quad b_\theta \triangleq \sum_{k=1}^{K} \beta_k S_{\theta,k} \left( r_{\theta,k} - \eta_{\theta,k} e \right). \tag{6}$$

For proving the convergence of the critic, we assume the policy is fixed. Thus, for each RB the induced MC is one for all the samples in this RB, so the sum over $N$ disappear for $A_\theta$ and $b_\theta$. Now we are ready to prove the following theorems, regarding Algorithm 1. We note that Theorems 2 and 3 state the critic and actor convergence.

**Theorem 2.** *(Convergence of the Critic to a fixed point)*
*Under Assumptions 1-6, for any given $\pi$ and $\{\eta_\tau\}, \{v_\tau\}$ as in the updates in Algorithm 1, we have $\eta_\tau \to \bar{\eta}_\theta$ and $v_\tau \to v^\pi$ with probability 1, where $v^\pi$ is obtained as a unique solution to $\Phi^\top A_\theta \Phi v + \Phi^\top b_\theta = 0$.*

The proof for Theorem 2 follows the proof for Lemma 5 in [6], see more details in the supplementary material A.2. For establishing the convergence of the actor updates, we define additional terms. Let $\mathcal{Z}$ denote the set of asymptotically stable equilibria of the ODE $\dot{\theta} = \hat{\Gamma}(-\nabla_\theta \bar{\eta}_\theta)$ and let $\mathcal{Z}^\epsilon$ be the $\epsilon$-neighborhood of $\mathcal{Z}$. Let $\bar{V}_k^{\pi_\theta}(\tilde{s}) = \sum_{\tilde{a}} \pi_\theta(\tilde{a}|\tilde{s}) \left( r(\tilde{s}, \tilde{a}) - \eta_{\theta,k} + \sum_{\tilde{s}'} P_k(\tilde{s}'|\tilde{s}, \tilde{a}) \phi(\tilde{s}')^\top v^{\pi_\theta} \right)$, and define

$$\xi^{\pi_\theta} = \sum_{k=1}^{K} \beta_k \sum_{\tilde{s}} \mu_{\theta,k}(\tilde{s}) \left( \phi(\tilde{s})^\top \nabla_\theta v^{\pi_\theta} - \nabla_\theta \bar{V}_k^{\pi_\theta}(\tilde{s}) \right).$$

**Theorem 3.** *(Convergence of the actor)*
*Under Assumptions 1-6, given $\epsilon > 0$, $\exists \delta > 0$ such that for $\theta_\tau$, $\tau \geq 0$ obtained using Algorithm 1, if $\sup_{\theta_\tau} \|\xi^{\pi_{\theta_\tau}}\| < \delta$, then $\theta_\tau \to \mathcal{Z}^\epsilon$ as $\tau \to \infty$ with probability one.*

The proof for Theorem 3 follows the proof for Theorem 2 in [6] and is given in the supplementary material A.3.

## 5.2 Sim2Real Asymptotic Convergence Properties

In this section we analyze the convergence properties of the Mixing Sim and Real algorithm we use. The main idea is that if sim and real are close in their dynamics through the MDP transition matrix many properties of their MDPs under the same policy are close as well. Moreover, we show that under the assumption of sim close to real, any process derived from both processes is close to both sim and real.

**Assumption 7.** *(Closeness of sim and real). For all $s, s' \in S$, $a \in A$, we have $|P_s(s'|s, a) - P_r(s'|s, a)| \leq \epsilon_{s2r}$.*

The following theorem states that if Assumption 7 holds then the convergence points of sim, real, and the mixed process (as defined in Algorithm 1) converge to close points.

**Theorem 4.** *Consider a policy $\pi_\theta(a|s)$ and Assumptions 1, 2, and 7. Then, for each $s, s' \in S$, $a \in A$, and $\forall \theta \in \Theta$ we have:*
*1. The induced MC of sim and real, $MC_s$ and $MC_r$, satisfy $|P_s^\theta(s'|s) - P_r^\theta(s'|s) \leq B_P \triangleq |A|\epsilon_{s2r}$.*
*2. Let $\tilde{P}_s^\theta \in \mathbb{R}^{(|S|-1)\times(|S|-1)}$ where its elements are identical to the first $(|S|-1) \times (|S|-1)$ elements of $P_s^\theta$. The corresponding stationary distributions satisfy $|\mu_s^\theta(s) - \mu_r^\theta(s)| \leq B_\mu \triangleq B_P|S|^3 \min_{\theta \in \Theta} \sqrt{SR_m^2}$, where $R_m$ is the largest eigenvalue of the matrix $\tilde{P}_s^\theta$.*
*3. The convergence points for the average reward and value functions under the policy for sim and real satisfy $\|\eta_s^\theta - \eta_r^\theta\| \leq B_\eta \triangleq B_\mu|S|$ and $\|\mathbf{v}_s^\theta - \mathbf{v}_r^\theta\| \leq B_\mu$.*

The proof for Theorem 4 is in the supplementary material B. Based on this Theorem, it follows immediately that any convex combination of "close" enough sim and real share the same properties as both sim and real. We defer to supplementary material the precise statement.

# 6 Experimental Evaluation

In this section we evaluate the performance of our proposed algorithm on two Fetch Push environments [37], one acts as the real environment and the other is the simulation environment [3]. Although our theoretical results are on the proposed mixing scheme with linear function approximation, in this section we focus on non-linear methodologies, i.e., using neural networks. We set $K = 2$ meaning there is only one real and one simulation environments. We denote by $q_r$ the probability of collecting samples from the real environment and by $\beta_r$ the probability of choosing samples from the real environment for the optimization process. We are interested in demonstrating the effect of different $q_r$ and $\beta_r$ values on the learning process. In addition, we investigate different mixing strategies for combining real and sim samples:

---

[3]The code for the experiments is available at: https://github.com/sdicastro/SimAndRealBetterTogether.

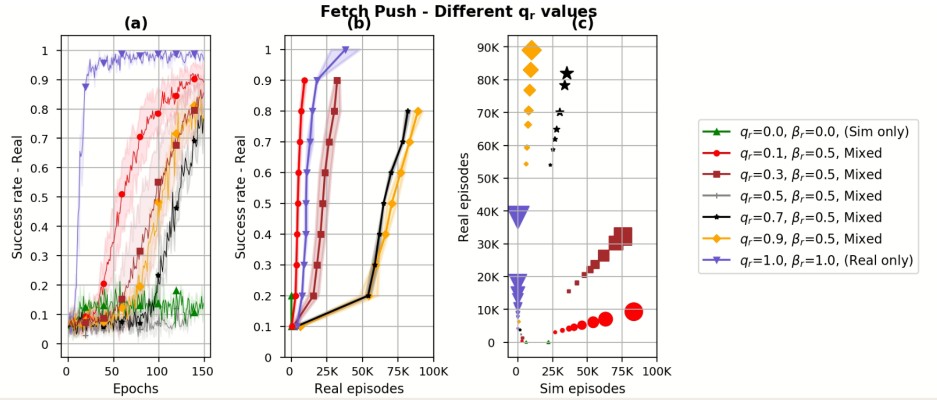

Figure 2: "Real only", "Sim only", and "Mixed" strategies with fixed $\beta_r$ and different $q_r$ values. **(a)** Success rate in the real environment vs. number of epochs. Each epoch corresponds to 100 episodes, mixed with real and sim episodes. The success rate is computed every epoch over 10 test episodes. **(b)** Success rate in the real environment vs. number of real episodes **(c)** Number of real episodes vs. number of sim episodes. The size of the markers corresponds to the increasing success rate. For all graphs, we repeated each experiment with 10 different random seeds and present the mean and standard deviation values.

1. "Mixed": real and sim episodes are collected according to Algorithm 1.

2. "Real only": The agent collects and optimize only real samples (i.e., $q_r = 1$ and $\beta_r = 1$).

3. "Sim only": The agent collects and optimize only sim samples (i.e., $q_r = 0$ and $\beta_r = 0$).

4. "Sim first": At the beginning, the agent collects and optimizes only sim samples. When the success rate in the sim environment reaches 0.7, we switch to sampling and optimizing only using real samples.

5. "Sim-dependent": At the beginning, the agent collects and optimizes only sim samples. When the success rate in the sim environment reaches 0.7, we switch to the "Mixed" strategy.

In the Fetch Push task, a robot arm needs to push an object on a table to a certain goal point. The state is represented by the gripper, object and target position and pose, as well as their velocities and angular velocities[4]. The action specifies the desired gripper position at the next time-step. The agent gets a reward of -1, if the desired goal was not yet achieved and 0 if it was achieved within some tolerance. To solve the task we used our mixing sim and real algorithm and replaced the linear actor-critic optimization scheme (lines 6-9 in Algorithm 1) with DDPG [29] together with Hindsight Experience Replay (HER; [1]) optimization scheme. We created the real and sim environments using the Mujoco simulator [46]. The difference between the environments is the friction between the object and the table. We preceded the following experiments with an experiment to depict a region of friction parameters where training the task using only sim samples and using the trained policy in the real environment does not solve the task (see supplementary material Section C.3).

We emphasize that we evaluate the performance in each experiment according to the success rate in the *real environment*, as this is the environment of final interest. In addition, we seek for mixing strategies that achieve the *lowest* number of *real* samples since usually they are costly and harder to get than sim samples.

**Different $q_r$ values**: We fix optimization parameter $\beta_r = 0.5$ and test different collection parameter $q_r = 0, 0.1, 0.3, 0.5, 0.7, 0.9, 1$. Results are presented in Figure 2. We notice that when the agent is trained using "Sim only" strategy ($q_r = 0$), it fails to solve the task in real (Figure 2a). Next, when the agent is trained using "Real only" strategy ($q_r = 1$), the task is solved. However, for achieving 0.9 success rate, "Real only" requires approximately 20K real episodes and to increase it to success rate of 1, it requires approximately 40K real episodes (Figures 2b and 2c). Observing the $q_r$ values in-between, we see that $q_r = 0.1$ achieves the best performance – it uses fewer ($\approx$ 10K) real episodes to achieve high success rates compared to the "Real only" strategy. Notice that as $q_r$ increases the

---

[4]The final dimension is 28 after removing non-informative dimensions.

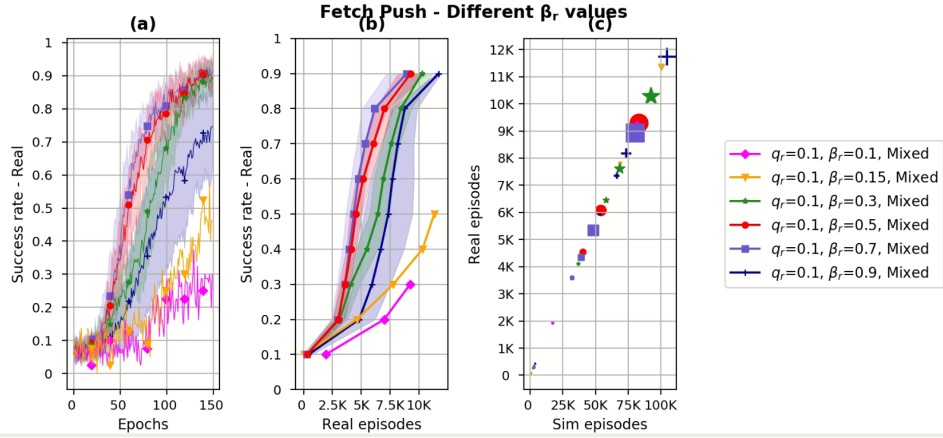

Figure 3: The "Mixed" strategy with fixed $q_r$ and different $\beta_r$ values. (a), (b) and (c) descriptions are the same as in Figure 2. In (c), the size of the markers corresponds to the increasing success rate: $0.1, 0.3, 0.5, 0.7, 0.9$.

performance deteriorates. This phenomenon can be explained due to the mixed samples distribution. When $q_r$ is low, most of the data distribution is based on sim, and real samples do not change it much, but only "fine tune" the learning. When $q_r$ increases, the data distribution is composed of two different environments which may confuse the agent.

**Different $\beta_r$ values**: In this experiment, we fix $q_r = 0.1$ and test for $\beta_r = 0.1, 0.15, 0.3, \dots 0.9$. Results are presented in Figure 3. When $\beta_r$ is low and equals $q_r$, the agent fails to solve the task (Figure 3a). However, when $\beta_r$ is higher than $q_r$, the performance improves where no significant differences are observed for $\beta_r = 0.3, 0.5, 0.7$. For $\beta_r = 0.7$, the algorithm achieves the best performance: high success rate of 0.9 while using fewer real episodes and fewer sim episodes compared to other $\beta_r$ values (Figures 3b and 3c). Interestingly, when $\beta_r$ is too high (with respect to $q_r$, i.e., $\beta_r = 0.9$) the performance deteriorates.

**Different Mixing Strategies:** We tested different mixing strategies. "Mixed", "Sim first" and "Sim-dependent" as described above. Results are presented in Figure 4. Using the "Sim-dependent" strategy reduced the required real and sim episodes to achieve 0.9 success rate comparing to the "Mixed" strategy with the same $q_r$ and $\beta_r$ values (Figure 4c). When using "Sim first" strategy, we observe that although in the beginning of the learning it uses only sim samples, once it switches to use only real samples, the agent requires many more real episodes to achieve success rate of 0.9 (compared to the "Mixed" and "Sim-dependent" strategies; Figures 4b and 4c). Although the most common approach is to train a policy in simulation and then use it as an initial starting point for the real system, we see that applying the mixing strategy after transferring the policy to real can reduce further the required real episodes while maintaining high success rate.

## 7 Conclusions and Future Work

In this work we analyzed a mixing strategy between simulation and real system samples. By separating the rate of collecting samples from each environment and the rate of choosing samples for the optimization process, we were able to achieve a significant reduction in the amount of real environment samples, comparing to the common strategy of using the same rate for both collection and optimization phases. This reduction is of special interest since usually the real samples are costly and harder to achieve. We believe this work can lead to a new line of research. First, finite sample analysis for our proposed algorithm can reveal its exact sample complexity. Comparing it to the sample complexity of learning only on real environment can emphasis the advantage of using the mixing strategy. Second, other replay buffer prioritization schemes can now be theoretically analyzed using the dynamics and properties of replay buffers we have developed. Third, our approach is limited to the online case, where new samples are collected during training. Adapting our approach to the offline case can discover new venues in the offline RL research. Fourth, learning the real samples collection rate and adapting it during training can further improve our approach.

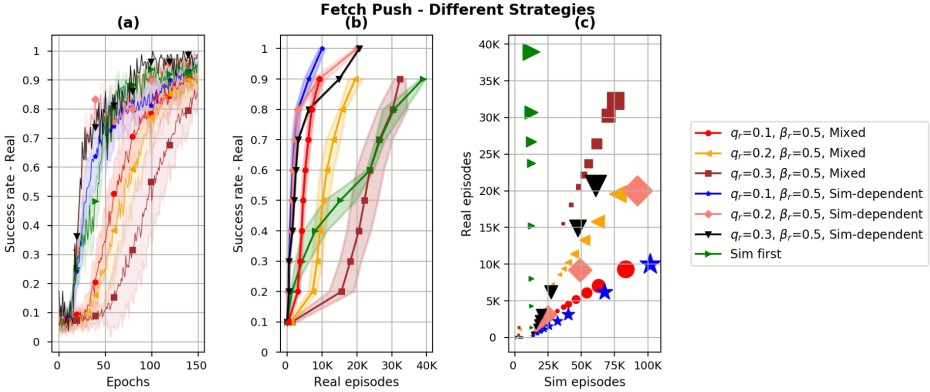

Figure 4: Comparing strategies: "Mixed", "Sim-dependent" and "Sim first". (a), (b) and (c) descriptions are the same as in Figure 2. It can be clearly seen in (c) that "Sim first" requires the most number of real episodes to achieve a high success rate. In addition, (b) and (c) demonstrate that for the same $(q_r, \beta_r)$ tuple, for example $(q_r = 0.2, \beta = 0.5)$, "Sim-dependent" strategy achieves higher success rates with less number of real episodes, compared to the "Mixing" strategy.

# 8 Acknowledgements

Research was conducted under ISF grant number 2199/20.

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
