# A   Proof of Main Lemmas and Theorems of Section 5.1

## A.1   Proof of Lemma 1

*Proof.* **1.** Proving Markovity requires that

$$P(Y_{\tau+1}|Y_\tau, Y_{\tau-1}, \ldots, Y_0) = P(Y_{\tau+1}|Y_\tau). \tag{7}$$

Let us denote $\mathcal{O}_{n_1}^{n_2} \triangleq \{\mathcal{O}_{n_1}, \ldots, \mathcal{O}_{n_2}\}$, $I_{n_1}^{n_2} \triangleq \{I_{n_1}, \ldots, I_{n_2}\}$ and $J_{n_1}^{n_2} \triangleq \{J_{n_1}, \ldots, J_{n_2}\}$. Recall that $\mathcal{O}_\tau = \{s_\tau, a_\tau, r_\tau, s_{\tau+1}\}$ and that the time index of entering a transition into RB(k) is $t(k, n) \in \{0, \ldots, \tau-1\}$ for all $k \in \{1, \ldots, K\}$ and for all $n \in \{1, \ldots N\}$. Index $n$ relates the position in RB(k) in which the transition is placed at time $\tau$. In addition, recall that $\mathcal{O}_{k,n} = \{s_{k,n}, a_{k,n}, r_{k,n}, s'_{k,n}\}$ where $s'_{k,n} \sim P_k(\cdot|s_{k,n})$. Let $RB_\tau(k)$ be RB(k) of MDP $M_k$ at time $\tau$, denoted as $RB_\tau(k) \triangleq \{\mathcal{O}_{k,1}, \ldots, \mathcal{O}_{k,N}\}_\tau \triangleq \mathcal{O}_{k,1}^{k,N}(\tau)$ .We denote the collection of all $RB_\tau(k)$ as $\bigcup\limits_{k=1}^{K} RB_\tau(k) \triangleq \bigcup\limits_{k=1}^{K} \mathcal{O}_{k,1}^{k,N}(\tau)$.

**Remark 1.** *Note that each time step that a transition enters some RB($\cdot$) is unique. That is, for a fixed $\tau$, $t(k, n) \neq t(k', n)$ for all $n$ and for $k \neq k'$. Moreover, $t(k, n) < t(k, n+1)$ for all $k$ and all $n$. In addition, note that when a new transition is pushed into the RB, the oldest transition in the RB is thrown away, and all the transitions in the RB, move one index forward, that is $\mathcal{O}_{k,n+1}(\tau+1) = \mathcal{O}_{k,n}(\tau)$ for $n = 1, \ldots N-1$ and $\mathcal{O}_{k,1}(\tau+1) = \mathcal{O}_\tau$.*

Computing the l.h.s. of (7) yields

$$
\begin{aligned}
P(Y_{\tau+1}|Y_\tau, Y_{\tau-1}, \ldots, Y_0) &\overset{1}{=} P\left( \bigcup_{k=1}^{K} RB_{\tau+1}(k), I_{\tau+1}, J_{\tau+1} \,\middle|\, \bigcup_{k=1}^{K} RB_\tau(k), I_\tau, J_\tau \ldots, \bigcup_{k=1}^{K} RB_0(k), I_0, J_0 \right) \\
&\overset{2}{=} P\left( \bigcup_{k=1}^{K} \mathcal{O}_{k,1}^{k,N}(\tau+1), I_{\tau+1}, J_{\tau+1} \,\middle|\, \bigcup_{k=1}^{K} \mathcal{O}_{k,1}^{k,N}(\tau), I_\tau, J_\tau \ldots, \bigcup_{k=1}^{K} \mathcal{O}_{k,1}^{k,N}(0), I_0, J_0 \right) \\
&\overset{3}{=} P\left( \bigcup_{k=1}^{K} \mathcal{O}_{k,1}^{k,N}(\tau+1), I_{\tau+1}, J_{\tau+1} \,\middle|\, I_0^\tau, J_0^\tau, \bigcup_{k=1}^{K} \mathcal{O}_{k,1}^{k,N}(\tau), \ldots, \bigcup_{k=1}^{K} \mathcal{O}_{k,1}^{k,N}(0) \right) \\
&\overset{4}{=} P\left( \bigcup_{k=1}^{K} \mathcal{O}_{k,1}^{k,N}(\tau+1) \,\middle|\, I_0^{\tau+1}, J_0^{\tau+1}, \bigcup_{k=1}^{K} \mathcal{O}_{k,1}^{k,N}(\tau), \ldots, \bigcup_{k=1}^{K} \mathcal{O}_{k,1}^{k,N}(0) \right) \\
&\quad \times P\left( I_{\tau+1} \,\middle|\, I_0^\tau, J_0^{\tau+1}, \bigcup_{k=1}^{K} \mathcal{O}_{k,1}^{k,N}(\tau), \ldots, \bigcup_{k=1}^{K} \mathcal{O}_{k,1}^{k,N}(0) \right) \\
&\quad \times P\left( J_{\tau+1} \,\middle|\, I_0^\tau, J_0^\tau, \bigcup_{k=1}^{K} \mathcal{O}_{k,1}^{k,N}(\tau), \ldots, \bigcup_{k=1}^{K} \mathcal{O}_{k,1}^{k,N}(0) \right) \\
&\overset{5}{=} P\left( \bigcup_{k=1}^{K} \mathcal{O}_{k,1}^{k,N}(\tau+1) \,\middle|\, \bigcup_{k=1}^{K} \mathcal{O}_{k,1}^{k,N}(\tau), I_\tau, J_\tau \right) \times P(I_{\tau+1}) \times P(J_{\tau+1}),
\end{aligned}
$$

where in equality (1) we use the definition, in equality (2) we wrote the RB samples explicitly, in equality (3) the terms were rearranged, in equality (4) we expressed the probability as a conditional product, and in equality (5) we use the fact that $I_\tau$ and $J_\tau$ are independent random variables and the rule of pushing transition $\mathcal{O}_\tau$ into RB($I_\tau$):

$$
\mathcal{O}_{t(k,1)}^{t(k,N)}(\tau+1) = \begin{cases} \mathcal{O}_{t(k,1)}^{t(k,N)}(\tau) & \text{if } k \neq I_\tau \\ \mathcal{O}_{t(k,1)}^{t(k,N-1)}(\tau) \cup \mathcal{O}_\tau & \text{if } k = I_\tau \end{cases}
$$

Similarly, computing the r.h.s of (7) yields

$$P(Y_{\tau+1}|Y_\tau) = P\left(\bigcup_{k=1}^K RB_{\tau+1}(k), I_{\tau+1}, J_{\tau+1}\middle|\bigcup_{k=1}^K RB_\tau(k), I_\tau, J_\tau\right)$$

$$= P\left(\bigcup_{k=1}^K RB_{\tau+1}(k)\middle|\bigcup_{k=1}^K RB_\tau(k), I_{\tau+1}, J_{\tau+1}, I_\tau, J_\tau\right)$$

$$\times P\left(I_{\tau+1}\middle|\bigcup_{k=1}^K RB_\tau(k), J_{\tau+1}, I_\tau, J_\tau\right) \times P\left(J_{\tau+1}\middle|\bigcup_{k=1}^K RB_\tau(k), I_\tau, J\tau\right)$$

$$= P\left(\bigcup_{k=1}^K RB_{\tau+1}(k)\middle|\bigcup_{k=1}^K RB_\tau(k), I_\tau, J_\tau\right) \times P(I_{\tau+1}) \times P(J_{\tau+1})$$

$$= P\left(\bigcup_{k=1}^K \mathcal{O}_{k,1}^{k,N}(\tau+1)\middle|\bigcup_{k=1}^K \mathcal{O}_{k,1}^{k,N}(\tau), I_\tau, J_\tau\right) \times P(I_{\tau+1}) \times P(J_{\tau+1}).$$

Both sides of (7) are equal and therefore $Y_\tau$ is Markovian.

**2.** According to Assumption 3, we assume that for every environment $k$ and for every policy $\pi$ the Markov Process induced by the MDP together with the policy $\pi$ is irreducible and aperiodic. In addition, we assume $\tau \geq \tau'$, where $\tau'$ is the time where we have full $K$ RBs, each one with $N$ transitions. This means that when a new transition arrives to RB(k), it requires throwing away the oldest transition in the buffer. We saw in part **1** that

$$P(Y_{\tau+1}|Y_\tau) = P\left(\bigcup_{k=1}^K \mathcal{O}_{k,1}^{k,N}(\tau+1)\middle|\bigcup_{k=1}^K \mathcal{O}_{k,1}^{k,N}(\tau), I_\tau, J_\tau\right) \times P(I_{\tau+1}) \times P(J_{\tau+1}). \quad (8)$$

Let $\mathbb{K} = \{1, \ldots K\}$ be an index set. We now write explicitly the following term

$$P\left(\bigcup_{k=1}^K \mathcal{O}_{k,1}^{k,N}(\tau+1)\middle|\bigcup_{k=1}^K \mathcal{O}_{k,1}^{k,N}(\tau), I_\tau, J_\tau\right)$$

$$= P\left(\mathcal{O}_{I_\tau,1}^{I_\tau,N}(\tau+1)\middle|\underbrace{\bigcup_{k\in\mathbb{K}\setminus I_\tau} \mathcal{O}_{k,1}^{k,N}(\tau+1), \bigcup_{k=1}^K \mathcal{O}_{k,1}^{k,N}(\tau), I_\tau, J_\tau}_{(a)}\right) \times P\left(\underbrace{\bigcup_{k\in\mathbb{K}\setminus I_\tau} \mathcal{O}_{k,1}^{k,N}(\tau+1)\middle|\bigcup_{k=1}^K \mathcal{O}_{k,1}^{k,N}(\tau), I_\tau, J_\tau}_{(b)}\right),$$

$$(9)$$

where we expressed the probability as a conditional product, separating RB($I_\tau$) at time $\tau+1$ from all other RB's. Note that in $(b)$: $\mathcal{O}_{k,1}^{k,N}(\tau+1) = \mathcal{O}_{k,1}^{k,N}(\tau)$ for all $k \neq I_\tau$ since these RB's do not change in this time-step.

We continue with expression (a).

$$P\left(\mathcal{O}_{I_\tau,1}^{I_\tau,N}(\tau+1)\middle|\bigcup_{k\in\mathbb{K}\setminus I_\tau} \mathcal{O}_{k,1}^{k,N}(\tau+1), \bigcup_{k=1}^K \mathcal{O}_{k,1}^{k,N}(\tau), I_\tau, J_\tau\right)$$

$$\overset{1}{=} P\left(\mathcal{O}_{I_\tau,1}^{I_\tau,N}(\tau+1)\middle|\mathcal{O}_{I_\tau,1}^{I_\tau,N}(\tau), I_\tau, J_\tau\right)$$

$$\overset{2}{=} P\left(\mathcal{O}_{I_\tau,1}(\tau+1)|\mathcal{O}_{I_\tau,1}(\tau), I_\tau, J_\tau\right)$$

$$\overset{3,(i=I_\tau,t_i=t_{I_\tau})}{=} P\left(s_{i,t_i}, a_{i,t_i}, r_{i,t_i}, s_{i,t_i+1}|s_{i,t_i-1}, a_{i,t_i-1}, r_{i,t_i-1}, s_{i,t_i}, I_\tau, J_\tau\right) \quad (10)$$

$$\overset{4}{=} P\left(a_{i,t_i}, r_{i,t_i}, s_{i,t_i+1}|s_{i,t_i}, I_\tau, J_\tau\right)$$

$$\overset{5}{=} P\left(r_{i,t_i}, s_{i,t_i+1}|s_{i,t_i}, a_{i,t_i}, I_\tau, J_\tau\right) \times P(a_{i,t_i}|s_{i,t_i}, I_\tau, J_\tau)$$

$$\overset{6}{=} P(s_{i,t_i+1}|s_{i,t_i}, a_{i,t_i}, r_{i,t_i}, I_\tau, J_\tau) \times P(r_{i,t_i}|s_{i,t_i}, a_{i,t_i}) \times \pi_{\theta(J_\tau)}(a_{i,t_i}|s_{i,t_i})$$

$$\overset{7}{=} P_{I_\tau}^{\pi_{\theta(J_\tau)}}(s_{i,t_i+1}|s_{i,t_i}) \times P(r_{i,t_i}|s_{i,t_i}, a_{i,t_i})$$

where in equality (1) we omitted all $RB_{\tau+1}(k)$ and $RB_\tau(k)$ for $k \neq I_\tau$ since they do not influence $RB_{\tau+1}(I_\tau)$ and for time $\tau$ we left only $RB_\tau(I_\tau)$. For equality (2) we recall Remark 1: $\mathcal{O}_{k,n+1}(\tau+1) = \mathcal{O}_{k,n}(\tau)$ for $n = 1, \ldots N-1$ and $\mathcal{O}_{k,1}(\tau+1) = \mathcal{O}_\tau$. Therefore, the transitions which are equal in both sides of the probability in equality (1), can be omitted. In equality (3) we write the transitions explicitly and change the notation for easier readability, $i = I_\tau$ and $t_i = t_{I_\tau}$ where $t_i$ is the last time-step in MDP $M_i$. In equality (4) we omit the condition of $s_{i,t_i}$ on itself and we keep the conditions only on $I_\tau$, $J_\tau$ and state $s_{i,t_i}$ since the process $\{s_{i,t_i}\}_{t_i \geq 0}$ is Markovian. In equalities (5) and (6) we express the probability as a conditional product. We denoted the policy $\pi_{\theta(J_\tau)}$ to emphasis that the policy depends on the observation sampled from $RB(J_\tau)$. Observe that in our setup, both $P(r_{i,t_i}|s_{i,t_i}, a_{i,t_i})$ and $\pi_{\theta(J_\tau)}(a_{i,t_i}|s_{i,t_i})$ are independent of $I_\tau$. Finally, in equality (7) we use the transitions in the induced Markov processes $\{s_{i,t_i}\}_{t_i \geq 0}$, $P^{\pi_\theta}_{I_\tau}(s_{i,t_i+1}|s_{i,t_i}) = P_{I_\tau}(s_{i,t_i+1}|s_{i,t_i}, a_{i,t_i}) \times \pi_\theta(a_{i,t_i}|s_{i,t_i})$ (recall that we used $i = I_\tau$).

Recall that $I_\tau$ and $J_\tau$ are independent random variables and that $P(I_{\tau+1} = k) = q_k$ and $P(J_{\tau+1} = k) = \beta_k$. Combining (8), (9) and (10) yields

$$P(Y_{\tau+1}|Y_\tau) = P^{\pi_{\theta(J_\tau)}}_{I_\tau}(s'|s) \times P(r|s,a) \times q_{I_{\tau+1}} \times \beta_{J_{\tau+1}}.$$

Using Assumption 3 and since the probability $P(r|s,a)$ does not influence the policy and MDP dynamics, the process $Y_\tau$ is aperiodic and irreducible.

$\square$

## A.2 Proof of Theorem 2

*Proof.* Recall that our TD-error update in line 6 in Algorithm 1 is defined as $\delta(\tilde{\mathcal{O}}^z) = \tilde{r}^z - \eta + \phi(\tilde{s}'^z)^\top v - \phi(\tilde{s}^z)^\top v$, where $\tilde{\mathcal{O}}^z = \{\tilde{s}^z, \tilde{a}^z, \tilde{r}^z, \tilde{s}'^z\}$. In the critic update in line 8 in Algorithm 1 we use an empirical mean over several sampled observations, denoted as $\{\tilde{\mathcal{O}}^z\}_{z=1}^{N_{\text{samples}}}$. Then, the critic update is defined as

$$v' = v + \alpha^v \frac{1}{N_{\text{samples}}} \sum_z \delta(\tilde{\mathcal{O}}^z)\phi(\tilde{s}^z).$$

Consider $N_{\text{samples}} = 1$. For a single sample update, we will use the following notations for the rest of the proof: $\tilde{\mathcal{O}} = \{\tilde{s}, \tilde{a}, \tilde{r}, \tilde{s}'\}$ and $\delta(\tilde{\mathcal{O}}) = \tilde{r} - \eta + \phi(\tilde{s}')^\top v - \phi(\tilde{s})^\top v$.

In this proof we follow the proof of Lemma 5 in [6]. Observe that the average reward and critic updates from Algorithm 1 can be written as

$$\eta_{\tau+1} = \eta_\tau + \alpha^\eta_\tau \left( F^\eta_\tau + M^\eta_{\tau+1} \right) \tag{11}$$
$$v_{\tau+1} = v_\tau + \alpha^v_\tau \left( F^v_\tau + M^v_{\tau+1} \right), \tag{12}$$

where

$$F^\eta_\tau \triangleq \mathbb{E}_{k \sim \beta, \tilde{\mathcal{O}} \sim \text{RB}(k), \tilde{s}, \tilde{a} \in \tilde{\mathcal{O}}} \left[ \tilde{r} - \eta | \mathcal{F}_\tau \right]$$
$$M^\eta_{\tau+1} \triangleq (\tilde{r} - \eta_\tau) - F^\eta_\tau$$
$$F^v_\tau \triangleq \mathbb{E}_{k \sim \beta, \tilde{\mathcal{O}} \sim \text{RB}(k), \tilde{s}, \tilde{a}, \tilde{s}' \in \tilde{\mathcal{O}}} \left[ \delta(\tilde{\mathcal{O}})\phi(\tilde{s}) \Big| \mathcal{F}_\tau \right]$$
$$M^v_{\tau+1} \triangleq \delta(\tilde{\mathcal{O}})\phi(\tilde{s}) - F^v_\tau$$

and $\mathcal{F}_\tau$ is a $\sigma$-algebra defined as $\mathcal{F}_\tau \triangleq \{\eta_t, v_t, M^\eta_t, M^v_t : t \leq \tau\}$.

We use Theorem 2.2 of [8] to prove convergence of these iterates. Briefly, this theorem states that given an iteration as in (11) and (12), these iterations are bounded w.p.1 if

**Assumption 8.** *1. $F^\eta_\tau$ and $F^v_\tau$ are Lipschitz, the functions $F_\infty(\eta) = \lim_{\sigma \to \infty} F^\eta(\sigma\eta)/\sigma$ and $F_\infty(v) = \lim_{\sigma \to \infty} F^v(\sigma v)/\sigma$ are Lipschitz, and $F_\infty(\eta)$ and $F_\infty(v)$ are asymptotically stable in the origin.*

*2. The sequences $M^\eta_{\tau+1}$ and $M^v_{\tau+1}$ are martingale difference noises and for some $C^\eta_0$, $C^v_0$*

$$\mathbb{E}\left[ (M^\eta_{\tau+1})^2 \big| \mathcal{F}_\tau \right] \leq C^\eta_0 (1 + \|\eta_\tau\|^2)$$
$$\mathbb{E}\left[ (M^v_{\tau+1})^2 \big| \mathcal{F}_\tau \right] \leq C^v_0 (1 + \|v_\tau\|^2).$$

We begin with the average reward update in (11). The ODE describing its asymptotic behavior corresponds to

$$\dot{\eta} = \mathbb{E}_{k\sim\beta, \tilde{\mathcal{O}}\sim\text{RB}(k), \tilde{s}, \tilde{a}\in\tilde{\mathcal{O}}}\left[\tilde{r} - \eta\right] \triangleq F^{\eta}. \tag{13}$$

$F^{\eta}$ is Lipschitz continuous in $\eta$. The function $F_{\infty}(\eta)$ exists and satisfies $F_{\infty}(\eta) = -\eta$. The origin is an asymptotically stable equilibrium for the ODE $\dot{\eta} = F_{\infty}(\eta)$ and the related Lyapunov function is given by $\eta^2/2$.

For the critic update, consider the ODE

$$\dot{v} = \mathbb{E}_{k\sim\beta, \tilde{\mathcal{O}}\sim\text{RB}(k), \tilde{s}, \tilde{a}, \tilde{s}'\in\tilde{\mathcal{O}}}\left[\delta(\tilde{\mathcal{O}})\phi(\tilde{s})\right] \triangleq F^{v}$$

In Lemma 5 we show that this ODE can be written as

$$\dot{v} = \Phi^{\top}A_{\theta}\Phi v + \Phi^{\top}b_{\theta}, \tag{14}$$

where $A_{\theta}$ and $b_{\theta}$ are defined in (6). $F^{v}$ is Lipschitz continuous in $v$ and $F_{\infty}(v)$ exists and satisfies $F_{\infty}(v) = \Phi^{\top}A_{\theta}\Phi v$. Consider the system

$$\dot{v} = F_{\infty}(v) \tag{15}$$

In assumption 5 we assume that $\Phi v \neq e$ for every $v \in \mathbb{R}^d$. Therefore, the only asymptotically stable equilibrium for (15) is the origin (see the explanation in the proof of Lemma 5 in [6]). Therefore, for all $\tau \geq 0$

$$\mathbb{E}\left[(M_{\tau+1}^{\eta})^2 \big| \mathcal{F}_{\tau}\right] \leq C_0^{\eta}(1 + \|\eta_{\tau}\|^2 + \|v_{\tau}\|^2)$$

$$\mathbb{E}\left[(M_{\tau+1}^{v})^2 \big| \mathcal{F}_{\tau}\right] \leq C_0^{v}(1 + \|\eta_{\tau}\|^2 + \|v_{\tau}\|^2)$$

for some $C_0^{\eta}, C_0^{v} < \infty$. $M_{\tau}^{\eta}$ can be directly seen to be uniformly bounded almost surely. Thus, Assumptions (A1) and (A2) of [8] are satisfied for the average reward, TD-error, and critic updates. From Theorem 2.1 of [8], the average reward, TD-error, and critic iterates are uniformly bounded with probability one. Note that when $\tau \to \infty$, (13) has $\bar{\eta}_{\theta}$ defined as in (3) as its unique globally asymptotically stable equilibrium with $V_2(\eta) = (\eta - \bar{\eta}_{\theta})^2$ serving as the associated Lyapunov function.

Next, suppose that $v = v^{\pi}$ is a solution to the system $\Phi^{\top}A_{\theta}\Phi v = 0$. Under Assumption 5, using the same arguments as in the proof of Lemma 5 in [6], $v^{\pi}$ is the unique globally asymptotically stable equilibrium of the ODE (14). Assumption 8 is now verified and under Assumption 6, the claim follows from Theorem 2.2, pp. 450 of [8].

$\square$

### A.2.1 Auxiliary Lemma for Theorem 2

The following Lemma computes the expectation of the critic update $\mathbb{E}\left[\delta(\tilde{\mathcal{O}})\phi(\tilde{s})\right]$.

**Lemma 5.** *Assume we have full $K$ RBs, each one with $N$ transitions. Then the following holds*

$$\mathbb{E}_{k\sim\beta, \tilde{\mathcal{O}}\sim RB(k), \tilde{s}, \tilde{a}, \tilde{s}'\in\tilde{\mathcal{O}}}\left[\delta(\tilde{\mathcal{O}})\phi(\tilde{s})\right] = \Phi^{\top}A_{\theta}\Phi v + \Phi^{\top}b_{\theta},$$

*where $A_{\theta}$ and $b_{\theta}$ are defined in (6).*

*Proof.* We note that due to the probabilistic nature of Algorithm 1, we do not know explicitly when each sample was pushed to any of the RBs. Next, we compute the expectation of the critic update with linear function approximation according to Algorithm 1.

$$
\begin{aligned}
&\mathbb{E}_{k\sim\beta,\tilde{\mathcal{O}}\sim\text{RB}(k),\tilde{s},\tilde{a},\tilde{s}'\in\tilde{\mathcal{O}}}\left[\delta(\tilde{\mathcal{O}})\phi(\tilde{s})\right]\\
&= \mathbb{E}_{k\sim\beta,\tilde{\mathcal{O}}\sim\text{RB}(k),\tilde{s},\tilde{a},\tilde{s}'\in\tilde{\mathcal{O}}}\left[\left(r(\tilde{s},\tilde{a})-\eta+\phi(\tilde{s}')^\top v-\phi(\tilde{s})^\top v\right)\phi(\tilde{s})\right]\\
&= \mathbb{E}_{k\sim\beta}\left[\mathbb{E}_{\tilde{\mathcal{O}}\sim\text{RB}(k),\tilde{s},\tilde{a},\tilde{s}'\in\tilde{\mathcal{O}}}\left[\left(r(\tilde{s},\tilde{a})-\eta+\phi(\tilde{s}')^\top v-\phi(\tilde{s})^\top v\right)\phi(\tilde{s})\right]\right]\\
&= \sum_{k=1}^{K}\beta_k\mathbb{E}_{\tilde{\mathcal{O}}\sim\text{RB}(k),\tilde{s},\tilde{a},\tilde{s}'\in\tilde{\mathcal{O}}}\left[\left(r(\tilde{s},\tilde{a})-\eta+\phi(\tilde{s}')^\top v-\phi(\tilde{s})^\top v\right)\phi(\tilde{s})\right]\\
&= \sum_{k=1}^{K}\beta_k\mathbb{E}_{\tilde{\mathcal{O}}\sim\text{RB}(k)}\left[\mathbb{E}_{\tilde{s},\tilde{a},\tilde{s}'\in\tilde{\mathcal{O}}_k}\left[\left(r(\tilde{s},\tilde{a})-\eta+\phi(\tilde{s}')^\top v-\phi(\tilde{s})^\top v\right)\phi(\tilde{s})\right]\right]\\
&= \sum_{k=1}^{K}\beta_k\sum_{n=1}^{N}\frac{1}{N}\mathbb{E}_{\tilde{s},\tilde{a},\tilde{s}'\in\tilde{\mathcal{O}}_{k,n}}\left[\left(r(\tilde{s},\tilde{a})-\eta+\phi(\tilde{s}')^\top v-\phi(\tilde{s})^\top v\right)\phi(\tilde{s})\right].
\end{aligned}
\tag{16}
$$

We note that in the last expression, the inner expectation is according to a tuple of indices $(k,n)$ where $k$ corresponds to $RB(k)$ and $n$ corresponds to the $n$-th transition in this $RB(k)$. Also, this sample $(k,n)$ corresponds to some $\theta_{t(k,n)}$. Recall that the time $t(k,n)$ is the time that the agent interacted with the $k$-th MDP and since then $n$ samples were added to $RB(k)$ (and the $n$ oldest samples were removed). In other words, sample $(k,n=1)$ is the newest sample in $RB(k)$ while sample $(k,n=N)$ is the oldest. Abusing notation, we define

$$
t(k,n) = \{t|\text{For the time }\tau,\text{ the time }t\text{ the }n\text{-th sample in }RB(k)\text{ was pushed.}\}
\tag{17}
$$

Next, we define the induced MC for the time $t(k,n)$ with a corresponding parameter $\theta_{t(k,n)}$. For this parameter, we denote the corresponding state distribution vector $\rho_{t(k,n)}$ and a transition matrix $P_{t(k,n)}$ (both induced by the policy $\pi_{\theta_{t(k,i)}}$. In addition, we define the following diagonal matrix $S_{t(k,n)} \triangleq \text{diag}(\rho_{t(k,n)})$. Similarly to [4] Lemma 6.5, pp.298, we can substitute the inner expectation

$$
\begin{aligned}
&\mathbb{E}_{\tilde{s},\tilde{a},\tilde{s}'\in\tilde{\mathcal{O}}_{k,n}}\left[\left(r(\tilde{s},\tilde{a})-\eta+\phi(\tilde{s}')^\top v-\phi(\tilde{s})^\top v\right)\phi(\tilde{s})\right] =\\
&\Phi^\top S_{t(k,n)}\left(P_{t(k,n)}-\mathbf{I}\right)\Phi v + \Phi^\top S_{t(k,n)}(r_{t(k,n)}-\eta_{\theta,k}e),
\end{aligned}
\tag{18}
$$

where $\mathbf{I}$ is the $|\mathcal{S}|\times|\mathcal{S}|$ identity matrix, $e$ in $|\mathcal{S}|\times 1$ vector of ones and $r_{k,n}$ is a $|\mathcal{S}|\times 1$ vector defined as $r_{t(k,n)}(s)=\sum_a\pi_{\theta_{t(k,n)}}(a|s)r(s,a)$. Combining equations (5), (16) and (18) yields

$$
\sum_{k=1}^{K}\sum_{n=1}^{N}\frac{\beta_k}{N}\left(\Phi^\top S_{t(k,n)}\left(P_{t(k,n)}-\mathbf{I}\right)\Phi v + \Phi^\top S_{t(k,n)}(r_{t(k,n)}-\eta_{\theta,k}e)\right) = \Phi^\top A_\tau\Phi v + \Phi^\top b_\tau,
\tag{19}
$$

In the limit, $\tau\to\infty$ and $\rho_{t(k,n)}\to\mu_{\theta,k}$ for all index $n$. Using $A_\theta$ and $b_\theta$ defined in (6), (16) can be expressed as

$$
\mathbb{E}_{k\sim\beta,\tilde{\mathcal{O}}\sim\text{RB}(k),\tilde{s},\tilde{a},\tilde{s}'\in\tilde{\mathcal{O}}}\left[\delta(\tilde{\mathcal{O}})\phi(\tilde{s})\right] = \Phi^\top A_\theta\Phi v + \Phi^\top b_\theta.
\tag{20}
$$

$\square$

## A.3 Proof of Theorem 3

*Proof.* Recall that our TD-error update in line 6 in Algorithm 1 is defined as $\delta(\tilde{\mathcal{O}}^z) = \tilde{r}^z - \eta + \phi(\tilde{s}'^z)^\top v - \phi(\tilde{s}^z)^\top v$, where $\tilde{\mathcal{O}}^z = \{\tilde{s}^z,\tilde{a}^z,\tilde{r}^z,\tilde{s}'^z\}$. In the actor update in line 9 in Algorithm 1 we use an empirical mean over several sampled observations, denoted as $\{\tilde{\mathcal{O}}^z\}_{z=1}^{N_{\text{samples}}}$. Then, the actor update is defined as

$$
\theta' = \Gamma\left(\theta - \alpha^\theta\frac{1}{N_{\text{samples}}}\sum_z\delta(\tilde{\mathcal{O}}^z)\nabla\log\pi_\theta(\tilde{a}^z|\tilde{s}^z)\right).
$$

Consider $N_{\text{samples}} = 1$. For a single sample update, we will use the following notations for the rest of the proof: $\tilde{\mathcal{O}} = \{\tilde{s}, \tilde{a}, \tilde{r}, \tilde{s}'\}$ and $\delta(\tilde{\mathcal{O}}) = \tilde{r} - \eta + \phi(\tilde{s}')^\top v - \phi(\tilde{s})^\top v$.

In this proof we follow the proof of Theorem 2 in [6]. Let $\delta^\pi(\tilde{\mathcal{O}}) = \tilde{r} - \eta + \phi(\tilde{s}')^\top v^\pi - \phi(\tilde{s})^\top v^\pi$, where $v^\pi$ is the convergent parameter of the critic recursion with probability one (see its definition in the proof for Theorem 2). Observe that the actor parameter update from Algorithm 1 can be written as

$$\theta_{\tau+1} = \Gamma\Big(\theta_\tau - \alpha_\tau^\theta\big(\delta(\tilde{\mathcal{O}})\nabla_\theta \log \pi_\theta(\tilde{a}|\tilde{s}) + F_\tau^\theta - F_\tau^\theta + N_\tau^{\theta_\tau} - N_\tau^{\theta_\tau}\big)\Big)$$
$$= \Gamma\Big(\theta_\tau - \alpha_\tau^\theta\big(M_{\tau+1}^\theta + (F_\tau^\theta - N_\tau^{\theta_\tau}) + N_\tau^{\theta_\tau}\big)\Big)$$

where

$$F_\tau^\theta \triangleq \mathbb{E}_{k\sim\beta, \tilde{\mathcal{O}}\sim\text{RB}(k), \tilde{a}, \tilde{s}, \tilde{s}'\in\tilde{\mathcal{O}}}\Big[\delta(\tilde{\mathcal{O}})\nabla_\theta \log \pi_\theta(\tilde{a}|\tilde{s})\Big|\mathcal{F}_\tau\Big]$$
$$M_{\tau+1}^\theta \triangleq \delta(\tilde{\mathcal{O}})\nabla_\theta \log \pi_\theta(\tilde{a}|\tilde{s}) - F_\tau^\theta$$
$$N_\tau^\theta \triangleq \mathbb{E}_{k\sim\beta, \tilde{\mathcal{O}}\sim\text{RB}(k), \tilde{s}, \tilde{a}, \tilde{s}'\in\tilde{\mathcal{O}}}\Big[\delta^{\pi_\theta}(\tilde{\mathcal{O}})\nabla_\theta \log \pi_\theta(\tilde{a}|\tilde{s})\Big|\mathcal{F}_\tau\Big]$$

and $\mathcal{F}_\tau$ is a $\sigma$-algebra defined as $\mathcal{F}_\tau \triangleq \{\eta_t, v_t, \theta_t, M_t^\eta, M_t^v, M_t^\theta : t \leq \tau\}$.

Since the critic converges along the faster timescale, from Theorem 2 it follows that $F_\tau^\theta - N_\tau^{\theta_\tau} = o(1)$. Now, let

$$M_2(\tau) = \sum_{r=0}^{\tau-1} \alpha_r^\theta M_{r+1}^\theta, \tau \geq 1.$$

The quantities $\delta(\tilde{\mathcal{O}})$ can be seen to be uniformly bounded since from the proof in Theorem 2, $\{\eta_\tau\}$ and $\{v_\tau\}$ are bounded sequences. Therefore, using Assumption 6, $\{M_2(\tau)\}$ is a convergent martingale sequence [5].

Consider the actor update along the slower timescale corresponding to $\alpha_\tau^\theta$ in line (9) in Algorithm 1. Let $v(\cdot)$ be a vector field on a set $\Theta$. Define another vector field: $\hat{\Gamma}\big(v(y)\big) = \lim_{0<\eta\to0}\Big(\frac{\Gamma\big(y+\eta v(y)\big)-y}{\eta}\Big)$. In case this limit is not unique, we let $\hat{\Gamma}\big(v(y)\big)$ be the set of all possible limit points (see pp. 191 of [27]). Consider now the ODE

$$\dot{\theta} = \hat{\Gamma}\Big(-\mathbb{E}_{k\sim\beta, \tilde{\mathcal{O}}\sim\text{RB}(k), \tilde{s}, \tilde{a}, \tilde{s}'\in\tilde{\mathcal{O}}}\Big[\delta^{\pi_\theta}(\tilde{\mathcal{O}})\nabla_\theta \log \pi_\theta(\tilde{a}|\tilde{s})\Big]\Big) \tag{21}$$

Substituting the result from Lemma 6, the above ODE is analogous to

$$\dot{\theta} = \hat{\Gamma}(-\nabla_\theta\bar{\eta}_\theta + \xi^{\pi_\theta}) = \hat{\Gamma}\big(-N_\tau^\theta\big) \tag{22}$$

where $\xi^{\pi_\theta} = \sum_{k=1}^K \beta_k \sum_{\tilde{s}} \mu_{\theta,k}(\tilde{s})\Big(\phi(\tilde{s})^\top\nabla_\theta v^{\pi_\theta} - \nabla_\theta\bar{V}_k^{\pi_\theta}(\tilde{s})\Big)$. Consider also an associated ODE:

$$\dot{\theta} = \hat{\Gamma}\big(-\nabla_\theta\bar{\eta}_\theta\big) \tag{23}$$

We now show that $h_1(\theta_\tau) \triangleq -N_\tau^{\theta_\tau}$ is Lipschitz continuous. Here $v^{\pi_{\theta_\tau}}$ corresponds to the weight vector to which the critic update converges along the faster timescale when the corresponding policy is $\pi_{\theta_\tau}$ (see Theorem 2). Note that $\mu_{\theta,k}(s), s \in \mathcal{S}, k \in \{1, \ldots, K\}$ are continuously differentiable in $\theta$ and have bounded derivatives. Also, $\bar{\eta}_{\theta_\tau}$ is continuously differentiable as well and has bounded derivative as can also be seen from (1). Further, $v^{\pi_{\theta_\tau}}$ can be seen to be continuously differentiable with bounded derivatives. Finally, $\nabla^2\pi_{\theta_\tau}(a|s)$ exists and is bounded. Thus $h_1(\theta_\tau)$ is a Lipschitz continuous function and the ODE (21) is well posed.

Let $\mathcal{Z}$ denote the set of asymptotically stable equilibria of (23) i.e., the local minima of $\bar{\eta}_\theta$, and let $\mathcal{Z}^\epsilon$ be the $\epsilon$-neighborhood of $\mathcal{Z}$. To complete the proof, we are left to show that as $\sup_\theta \|\xi^{\pi_\theta}\| \to 0$ (viz. $\delta \to 0$), the trajectories of (22) converge to those of (23) uniformly on compacts for the same initial condition in both. This claim follows the same arguments as in the proof of Theorem 2 in [6].

$\square$

### A.3.1 Auxiliary Lemma for Theorem 3

The following Lemma computes the expectation of $\mathbb{E}\left[\delta^{\pi_\theta}(\tilde{\mathcal{O}})\nabla_\theta \log \pi_\theta(\tilde{a}|\tilde{s})\right]$.

**Lemma 6.** *Assume we have full $K$ RBs, each one with $N$ transitions. Then the following holds*

$$\mathbb{E}_{k\sim\beta,\tilde{\mathcal{O}}\sim RB(k),\tilde{s},\tilde{a},\tilde{s}'\in\tilde{\mathcal{O}}}\left[\delta^{\pi_\theta}(\tilde{\mathcal{O}})\nabla_\theta \log \pi_\theta(\tilde{a}|\tilde{s})\right]$$

$$= \nabla_\theta \bar{\eta}_\theta - \sum_{k=1}^{K}\beta_k \sum_{\tilde{s}} \mu_{\theta,k}(\tilde{s})\left(\phi(\tilde{s})^\top \nabla_\theta v^{\pi_\theta} - \nabla_\theta \bar{V}_k^{\pi_\theta}(\tilde{s})\right),$$

*where $\bar{V}_k^{\pi_\theta}(\tilde{s}) = \sum_{\tilde{a}\in\mathcal{A}} \pi_\theta(\tilde{a}|\tilde{s})\left(r(\tilde{s},\tilde{a}) - \eta_{\theta,k} + \sum_{\tilde{s}'\in\mathcal{S}} P_k(\tilde{s}'|\tilde{s},\tilde{a})\phi(\tilde{s}')^\top v^{\pi_\theta}\right).$*

*Proof.* We compute the expectation of $\delta^{\pi_\theta}(\tilde{\mathcal{O}})\nabla_\theta \log \pi_\theta(\tilde{a}|\tilde{s})$ with linear function approximation according to Algorithm 1. Due to the probabilistic nature of Algorithm 1, we do not know explicitly when each transition was pushed to any of the RBs. Recall that the tuple $(k,n)$ corresponds to some $\theta_{t(k,n)}$ where time $t(k,n)$ was defined in Section 4. We use the same notations for the state distribution vector $\rho_{t(k,n)}$ and a transition matrix $P_{t(k,n)}$ (both induced by the policy $\pi_{\theta_{t(k,n)}}$, as in the proof for Lemma 5). We define now the following term:

$$\bar{V}_k^{\pi_{\theta_{t(k,n)}}}(\tilde{s}) = \sum_{\tilde{a}\in\mathcal{A}} \pi_{\theta_{t(k,n)}}(\tilde{a}|\tilde{s})\bar{Q}_k^{\pi_{\theta_{t(k,n)}}}(\tilde{s},\tilde{a}) = \sum_{\tilde{a}\in\mathcal{A}} \pi_{\theta_{t(k,n)}}(\tilde{a}|\tilde{s})\left(r(\tilde{s},\tilde{a}) - \eta_{\theta,k} + \sum_{\tilde{s}'\in\mathcal{S}} P_k(\tilde{s}'|\tilde{s},\tilde{a})\phi(\tilde{s}')^\top v^{\pi_\theta}\right),$$

$$(24)$$

where $\bar{V}_k^{\pi_{\theta_{t(k,n)}}}(\tilde{s})$ and $\bar{Q}_k^{\pi_{\theta_{t(k,n)}}}(\tilde{s},\tilde{a})$ correspond to policy $\pi_{\theta_{t(k,n)}}$. Note that here, the convergent critic parameter $v^{\pi_\theta}$ is used. Let's look at the gradient of (24):

$$\nabla_\theta \bar{V}_k^{\pi_{\theta_{t(k,n)}}}(\tilde{s}) = \nabla_\theta \left(\sum_{\tilde{a}\in\mathcal{A}} \pi_{\theta_{t(k,n)}}(\tilde{a}|\tilde{s})\bar{Q}_k^{\pi_{\theta_{t(k,n)}}}(\tilde{s},\tilde{a})\right)$$

$$= \sum_{\tilde{a}\in\mathcal{A}} \nabla_\theta \pi_{\theta_{t(k,n)}}(\tilde{a}|\tilde{s})\left(r(\tilde{s},\tilde{a}) - \eta_{\theta,k} + \sum_{\tilde{s}'\in\mathcal{S}} P_k(\tilde{s}'|\tilde{s},\tilde{a})\phi(\tilde{s}')^\top v^{\pi_\theta}\right)$$

$$+ \sum_{\tilde{a}\in\mathcal{A}} \pi_{\theta_{t(k,n)}}(\tilde{a}|\tilde{s})\left(-\nabla_\theta \eta_{\theta,k} + \sum_{\tilde{s}'\in\mathcal{S}} P_k(\tilde{s}'|\tilde{s},\tilde{a})\phi(\tilde{s}')^\top \nabla_\theta v^{\pi_\theta}\right)$$

$$= \sum_{\tilde{a}\in\mathcal{A}} \nabla_\theta \pi_{\theta_{t(k,n)}}(\tilde{a}|\tilde{s})\left(r(\tilde{s},\tilde{a}) - \eta_{\theta,k} + \sum_{\tilde{s}'\in\mathcal{S}} P_k(\tilde{s}'|\tilde{s},\tilde{a})\phi(\tilde{s}')^\top v^{\pi_\theta}\right)$$

$$- \nabla_\theta \eta_{\theta,k} + \sum_{\tilde{a}\in\mathcal{A}} \pi_{\theta_{t(k,n)}}(\tilde{a}|\tilde{s})\sum_{\tilde{s}'\in\mathcal{S}} P_k(\tilde{s}'|\tilde{s},\tilde{a})\phi(\tilde{s}')^\top \nabla_\theta v^{\pi_\theta}$$

where with abuse of notation, $\nabla_\theta \pi_{\theta_{t(k,n)}}(\tilde{a}|\tilde{s}) = \nabla_\theta \pi_{\theta|\theta=\theta_{t(k,n)}}(\tilde{a}|\tilde{s})$. In the limit, $\tau \to \infty$ and $\rho_{t(k,n)} \to \mu_{\theta,k}$ for all index $n$. Summing both sides over $\beta$ distribution and stationary distribution $\mu_{\theta,k}$

$$\sum_{k=1}^{K}\beta_k \sum_{\tilde{s}} \mu_{\theta,k}(\tilde{s})\nabla_\theta \bar{V}_k^{\pi_\theta}(\tilde{s})$$

$$= \sum_{k=1}^{K}\beta_k \sum_{\tilde{s}} \mu_{\theta,k}(\tilde{s})\sum_{\tilde{a}\in\mathcal{A}} \nabla_\theta \pi_\theta(\tilde{a}|\tilde{s})\left(r(\tilde{s},\tilde{a}) - \eta_{\theta,k} + \sum_{\tilde{s}'\in\mathcal{S}} P_k(\tilde{s}'|\tilde{s},\tilde{a})\phi(\tilde{s}')^\top v^{\pi_\theta}\right)$$

$$+ \sum_{k=1}^{K}\beta_k \sum_{\tilde{s}} \mu_{\theta,k}(\tilde{s})\left(-\nabla_\theta \eta_{\theta,k} + \sum_{\tilde{a}\in\mathcal{A}} \pi_\theta(\tilde{a}|\tilde{s})\sum_{\tilde{s}'\in\mathcal{S}} P_k(\tilde{s}'|\tilde{s},\tilde{a})\phi(\tilde{s}')^\top \nabla_\theta v^{\pi_\theta}\right)$$

$$= \mathbb{E}_{k\sim\beta,\tilde{\mathcal{O}}\sim RB(k),\tilde{s},\tilde{a},\tilde{s}'\in\tilde{\mathcal{O}}}\left[\delta^{\pi_\theta}(\tilde{\mathcal{O}})\nabla_\theta \log \pi_\theta(\tilde{a}|\tilde{s})\right]$$

$$- \nabla_\theta \bar{\eta}_\theta + \sum_{k=1}^{K}\beta_k \sum_{\tilde{s}} \mu_{\theta,k}(\tilde{s})\sum_{\tilde{a}\in\mathcal{A}} \pi_\theta(\tilde{a}|\tilde{s})\sum_{\tilde{s}'\in\mathcal{S}} P_k(\tilde{s}'|\tilde{s},\tilde{a})\phi(\tilde{s}')^\top \nabla_\theta v^{\pi_\theta}$$

We will write in short $\mathbb{E}\left[\delta^{\pi_\theta}(\tilde{\mathcal{O}})\nabla_\theta \log \pi_\theta(\tilde{a}|\tilde{s})\right] = \mathbb{E}_{k\sim\beta,\tilde{\mathcal{O}}\sim\mathrm{RB}(k),\tilde{s},\tilde{a},\tilde{s}'\in\tilde{\mathcal{O}}}\left[\delta^{\pi_\theta}(\tilde{\mathcal{O}})\nabla_\theta \log \pi_{\theta_{t(k,i)}}(\tilde{a}|\tilde{s})\right]$.
Then:

$$\nabla_\theta \bar{\eta}_\theta = \mathbb{E}\left[\delta^{\pi_\theta}(\tilde{\mathcal{O}})\nabla_\theta \log \pi_\theta(\tilde{a}|\tilde{s})\right]$$
$$+ \sum_{k=1}^{K}\beta_k \sum_{\tilde{s}}\mu_{\theta,k}(\tilde{s})\left(\sum_{\tilde{a}\in\mathcal{A}}\pi_\theta(\tilde{a}|\tilde{s})\sum_{\tilde{s}'\in\mathcal{S}}P_k(\tilde{s}'|\tilde{s},\tilde{a})\phi(\tilde{s}')^\top\nabla_\theta v^{\pi_\theta} - \nabla_\theta\bar{V}_k^{\pi_\theta}(\tilde{s})\right).$$

Since $\mu_{\theta,k}$ is the stationary distribution for each environment $k$,

$$\sum_{\tilde{s}}\mu_{\theta,k}(\tilde{s})\sum_{\tilde{a}\in\mathcal{A}}\pi_\theta(\tilde{a}|\tilde{s})\sum_{\tilde{s}'\in\mathcal{S}}P_k(\tilde{s}'|\tilde{s},\tilde{a})\phi(\tilde{s}')^\top\nabla_\theta v^{\pi_\theta} = \sum_{\tilde{s}}\mu_{\theta,k}(\tilde{s})\sum_{\tilde{s}'\in\mathcal{S}}P_{\theta,k}(\tilde{s}'|\tilde{s})\phi(\tilde{s}')^\top\nabla_\theta v^{\pi_\theta}$$
$$= \sum_{\tilde{s}'}\sum_{\tilde{s}}\mu_{\theta,k}(\tilde{s})P_{\theta,k}(\tilde{s}'|\tilde{s})\phi(\tilde{s}')^\top\nabla_\theta v^{\pi_\theta}$$
$$= \sum_{\tilde{s}'}\mu_{\theta,k}(\tilde{s}')\phi(\tilde{s}')^\top\nabla_\theta v^{\pi_\theta},$$

Then,

$$\nabla_\theta \bar{\eta}_\theta = \mathbb{E}\left[\delta^{\pi_\theta}(\tilde{\mathcal{O}})\nabla_\theta \log \pi_\theta(\tilde{a}|\tilde{s})\right] + \sum_{k=1}^{K}\beta_k \sum_{\tilde{s}}\mu_{\theta,k}(\tilde{s})\left(\phi(\tilde{s})^\top\nabla_\theta v^{\pi_\theta} - \nabla_\theta\bar{V}_k^{\pi_\theta}(\tilde{s})\right)$$

The result follows immediately.

$\square$

# B  Proof of Main Lemmas and Theorems of Section 5.2

## B.1  Proof of Theorem 4

*Proof.* **1.** We have a common policy to both sim and real. Thus,

$$\left|P_s^\theta(s'|s) - P_r^\theta(s'|s)\right| = \left|\sum_{a\in A}P_s(s'|s,a)\pi_\theta(a|s) - \sum_{a\in A}P_r(s'|s,a)\pi_\theta(a|s)\right|$$
$$= \left|\sum_{a\in A}\pi_\theta(a|s)\left(P_s(s'|s,a) - P_r(s'|s,a)\right)\right| \tag{25}$$
$$\leq \sum_{a\in A}\pi_\theta(a|s)\left|P_s(s'|s,a) - P_r(s'|s,a)\right|$$
$$\leq |A|\epsilon_{\mathrm{s2r}},$$

where the last inequality is due to Assumption 7.

**2.** The stationary distribution satisfies $\mu_s^{\theta\top}P_s^\theta = \mu_s^{\theta\top}$. Let us define $\Delta P \triangleq P_s^\theta - P_r^\theta$ and $\Delta\mu \triangleq \mu_s^\theta - \mu_r^\theta$. Then, we have

$$\mu_s^{\theta\top}(P_s^\theta - I) = 0$$
$$(\mu_r^\theta + \Delta\mu)^\top(P_s^\theta - I) = 0$$
$$\mu_r^{\theta\top}(P_s^\theta - I) + \Delta\mu^\top(P_s^\theta - I) = 0$$
$$\mu_r^{\theta\top}(\Delta P + P_r^\theta - I) + \Delta\mu^\top(P_s^\theta - I) = 0 \tag{26}$$
$$\mu_r^{\theta\top}(P_r^\theta - I) + \mu_r^{\theta\top}\Delta P + \Delta\mu\left(P_s^\theta - I\right) = 0$$
$$\mu_r^{\theta\top}\Delta P + \Delta\mu\left(P_s^\theta - I\right) = 0$$
$$\Delta\mu\left(P_s^\theta - I\right) = -\mu_r^{\theta\top}\Delta P.$$

We note that since $P_s^\theta$ satisfies Assumption 2, it is of degree $|S| - 1$ ($P_s^\theta$ has only one eigenvalue equals 1, thus, $I - P_s^\theta$ has only one eigenvalue equals 0). Without loss of generality, we define $\tilde{\Delta\mu}$ to be a vector with the first $|S| - 1$ entries, $\tilde{P}_s^\theta$ to be a sub-matrix with the first $(|S| - 1) \times (|S| - 1)$ entries of $P_s^\theta$, $\tilde{\mu}_r^\theta$ a vector with the first $|S| - 1$ first entries of $\mu_r^\theta$, $\tilde{\Delta P}$ a sub-matrix with the first $(|S| - 1) \times (|S| - 1)$ entries of $\Delta P$, and $\tilde{I}$ to be identity matrix of dimension $S - 1$. As a result, we have the following full rank equations system:

$$\tilde{\Delta\mu}\left(\tilde{P}_s^\theta - I\right) = -\tilde{\mu}_r^{\theta\top}\tilde{\Delta P}, \tag{27}$$

and the matrix $\left(\tilde{P}_s^\theta - I\right)$ is of full rank and invertible. Thus,

$$\tilde{\Delta\mu} = -\tilde{\mu}_r^{\theta\top}\tilde{\Delta P}\left(\tilde{P}_s^\theta - I\right)^{-1}. \tag{28}$$

We apply the Frobenius norm on both sides and get

$$\begin{aligned}
\|\tilde{\Delta\mu}\|_F &= \left\|\tilde{\mu}_r^{\theta\top}\tilde{\Delta P}\left(\tilde{P}_s^\theta - I\right)^{-1}\right\|_F. \\
&\leq \left\|\tilde{\mu}_r^\theta\right\|_F\left\|\tilde{\Delta P}\right\|_F\left\|\left(\tilde{P}_s^\theta - I\right)^{-1}\right\|_F. \\
&\leq 1 \cdot |S|^2 \cdot \epsilon\left\|\left(\tilde{P}_s^\theta - I\right)^{-1}\right\|_F.
\end{aligned} \tag{29}$$

We note that according to Assumption 1, $\Theta$ is compact and according to Assumption 3 the induced MC is aperiodic and irreducible. Therefore, the Frobenius norm of the latter norm (for all $\theta \in \Theta$) gets both the maximum and the minimum in $\Theta$. Using Gersgorin Theorem ([17]; Theorem 6.1.1) on matrix $\tilde{P}_s^\theta - I$, and since the diagonal is greater than 1 and for each row, the off diagonal entries sum to less than 1, all the eigenvalues of $\tilde{P}_s^\theta - I$ are strictly above some value $R_G > 0$. As a result, the eigenvalues of $(\tilde{P}_s^\theta - I)^{-1}$ are bounded by $R_G^{-1}$. Using Assumption 3, the matrix $(\tilde{P}_s^\theta)^{-1}$ is bounded for all $\theta \in \Theta$ by $R_M \triangleq \max \theta \in \Theta R_G^{-1}$, and the Frobenius of the latter norm is bounded by $\sqrt{S \cdot R_M}$. Summarizing, $\|\tilde{\Delta\mu}\| \leq \epsilon|S|^2 \min_{\theta \in \Theta}\sqrt{SR_m^2}$ for the first $S - 1$ states. We left with proving that for the last state, the same hold. Since $\sum_{i=1}^{|S|}\mu_r^\theta(i) = 1$ and $\sum_{i=1}^{|S|}\mu_s^\theta(i) = 1$, subtracting these two equations and rearranging yield

$$\mu_s^\theta(|S|) - \mu_r^\theta(|S|) = -\sum_{i=1}^{|S|-1}\left(\mu_s^\theta(i) - \mu_r^\theta(i)\right).$$

Applying Frobenius norm yields the desired result.

**3.** The boundedness of the average reward is immediate from part 2, i.e.,

$$\begin{aligned}
\|\eta_s^\theta - \eta_r^\theta\|_F &= \|\mu_r^{\theta\top}r - \mu_s^{\theta\top}r\|_F \\
&\leq \|\mu_r^{\theta\top} - \mu_s^{\theta\top}\|_F \cdot \|r\|_F \\
&\leq B_\mu|S|.
\end{aligned} \tag{30}$$

Similarly to part 2, the value function for sim and real are

$$\begin{aligned}
\mathbf{v}_s^\theta &= r - \eta + P_s^\theta\mathbf{v}_s^\theta, \\
\mathbf{v}_r^\theta &= r - \eta + P_r^\theta\mathbf{v}_r^\theta.
\end{aligned} \tag{31}$$

Subtracting both yields

$$\mathbf{v}_s^\theta - \mathbf{v}_r^\theta = P_s^\theta\mathbf{v}_s^\theta - P_r^\theta\mathbf{v}_r^\theta. \tag{32}$$

We add and subtract $P_s^\theta\mathbf{v}_r^\theta$ and rearrange to get

$$\begin{aligned}
\mathbf{v}_s^\theta - \mathbf{v}_r^\theta &= P_s^\theta\mathbf{v}_s^\theta - P_s^\theta\mathbf{v}_r^\theta + P_s^\theta\mathbf{v}_r^\theta - P_r^\theta\mathbf{v}_r^\theta \\
\mathbf{v}_s^\theta - \mathbf{v}_r^\theta &= P_s^\theta(\mathbf{v}_s^\theta - \mathbf{v}_r^\theta) + (P_s^\theta - P_r^\theta)\mathbf{v}_r^\theta \\
(I - P_s^\theta)(\mathbf{v}_s^\theta - \mathbf{v}_r^\theta) &= (P_s^\theta - P_r^\theta)\mathbf{v}_r^\theta.
\end{aligned} \tag{33}$$

Similarly to 2, we have an under-determined equation system. We assume that for both BEs of $\mathbf{v}_s^\theta(s^*) = \mathbf{v}_r^\theta(s^*) = 0$ in order for (31) to be each with a unique solution. Now, similarly to 2, we look at the $|S| - 1$ first equations (that now has a unique solution)

$$(\tilde{I} - \tilde{P}_s^\theta)(\tilde{\mathbf{v}}_s^\theta - \tilde{\mathbf{v}}_r^\theta) = (\tilde{P}_s^\theta - \tilde{P}_r^\theta)\tilde{\mathbf{v}}_r^\theta. \tag{34}$$

Again, similarly to 2 we get the desired result.

$\square$

## B.2 Corollary for Theorem 4

The following corollary follows immediately from Theorem 4 and establishes that any convex combination of "close" enough sim and real share the same properties as both sim and real.

**Corollary 7.** *Assuming the same as in Theorem 4, if $Y$ is a process where its dynamics can be described as $P_Y = \beta P_s(s'|s,a) + (1-\beta)P_r(s'|s,a)$ for $0 \le \beta \le 1$, then $Y$ satisfies the same properties as of Theorem 4 w.r.t. the real process.*

*Proof.* The process $Y$ is a convex combination of both sim and real, therefore, the distance between $P_r$ and $P_Y$ is smaller then the distance between $P_r$ and $P_s$. Using Theorem 4 the result follows immediately. $\square$

## C  Experiment Details of Section 6

We trained the Fetch Push task using the DDPG algorithm [29] together with HER [1]. For DDPG, HER and FetchPush task we used the same hyper-parameters as in [1]. For completeness, we specify the hyper-parameters and task parameters used in our experiments.

### C.1  Training procedure

We train for 150 epochs. Each epoch consists of 50 cycles where each cycle consists of running the policy for 2 episodes per worker. Every episode consists of 50 environment time-steps. Then, 40 optimization steps are performed on mini-batches of size 256 sampled uniformly from a replay buffer consisting of $10^6$ transitions. For improved efficiency, the whole training procedure is distributed over 8 threads (workers) which average the parameters after every update. Training for 150 epochs took us approximately 2h using 8 cpu cores. The networks are optimized using the Adam optimizer [24] with learning rate of 0.001. We update the target networks after every cycle using the decay coefficient of 0.95. We use the discount factor of $\gamma = 0.98$ for all transitions and we clip the targets used to train the critic to the range of possible values, i.e. $[-\frac{1}{1-\gamma}, 0]$. The behavioral policy we use for exploration works as follows. With probability 0.3 we sample (uniformly) a random action from the hypercube of valid actions. Otherwise, we take the output of the policy network and add independently to every coordinate normal noise with standard deviation equal to 0.2 of the total range of allowed values on this coordinate. Goals selection for HER algorithm was performed using the "future" HER strategy with $k = 4$. See [1] for additional details.

### C.2  Networks architecture

The architecture of the actor and critic networks is a Multi-Layer Perceptrons (MLP) with 3 hidden layers and ReLu activation function. Each layer has 256 hidden units. The actor output layer uses the tanh activation function and is rescaled so that it lies in the range $[-5\text{cm}, 5\text{cm}]$. We added the $L_2$ norm of the actions to the actor loss function to prevent tanh saturation and vanishing gradients (in the same way as in [1]). We rescale the inputs to the critic and actor networks so that they have mean zero and standard deviation equal to one and then clip them to the range $[-5, 5]$. Means and standard deviations used for rescaling are computed using all the observations encountered so far in the training.

### C.3  Task parameters

The initial position of the gripper is fixed, located 20cm above the table. The initial position of the box on the table is randomized, in the 30cm $\times$ 30cm square with the center directly under the gripper.

The width of the box is 5cm. The goal position is sampled uniformly from the same square as the box position.

The state space is 28-dimensional: 25 dimensions for the gripper and box poses and velocities and 3 for the goal position. The action space is 4-dimensional. Three dimensions specify the desired relative gripper position at the next time-step. The last dimension specifies the desired distance between the 2 fingers which are position controlled. Our task does not require gripper rotation and therefore we keep it fixed.

## C.4 Friction values

In our experiments, the difference between the real and sim environments is the friction between the box and the table. The friction parameter is a vector of 5 dimensions: two tangential, one torsional, two rolling. In our experiments, the friction values in the real environment are $[0.03, 1., 0.005, 0.0001, 0.0001]$ and the friction values in the sim environment are $[2., 2., 0.005, 0.01, 0.0001]$. The sim friction values were chosen after a preceding experiment on the friction range $[1.8, 2.2] \times [1.8, 2.2] \times [0.005] \times [0.0001, 0.1] \times [0.0001, 0.1]$, respectively, which ensured that if we train a policy only on the simulator and use this policy in the real environment, it does not solve the task. In this way we have a simulator which is close to the real world, but not identical to it, what usually happens when designing simulators for real systems such as a robotic arm.

## C.5 Performance evaluation

In our experiments, for each environment (real or sim) the task is solved if in the last time-step of an episode, the box position satisfies $\|$box position - goal position$\|_2 \leq 0.05$. After each training epoch, we tested in the *real* environment the trained policy for 10 episodes. The test was performed separately for each one of the 8 workers. For calculating the final success rate for each epoch we averaged the local success rate form each worker. Finally, for each $q_r$ and $\beta_r$ values, and for each mixing strategy, we repeated the experiment with 10 different random seeds.