# OpenReview forum: "Sim and Real: Better Together"
_NeurIPS.cc/2021/Conference — NeurIPS 2021 Poster_

### Official Review · Reviewer_rbtG · 2021-07-16

**Rating:** 4
**Confidence:** 3

**Summary:**

The paper addresses the challenge of sim-to-real transfer, i.e., how to use simulations for robot learning in the real world. It proposes the use of information from rollouts in both the simulated and the real world. The underlying technique is reminiscent of importance sampling. While the approach is explained for transfer between two environments (sim and real) it is applicable to any situation that involves K environments. At a basic level, the approach keeps track of K individual Replay Buffers which are used for RL. An agent chooses with some probability an environment and then with yet another probability samples a replay buffer to update the policy. In effect, the data from both the simulated and the real environment are used for training albeit with different mixing probabilities. The core contribution (in my opinion) of this paper is a theoretical proof that guarantees convergence of this mixing scheme. The proof shows that sampling transitions is a Markov process and that under some (reasonably benign) conditions it converges to a fix point. The approach is rigorous and the appendix includes all the individual steps of the proof. Two critical assumptions that are made in the proof is that the real and simulated transitions are close to each other (defined as difference of probabilities is below a threshold). In addition, the current proof assumes a linear approximation of the value function. The approach is evaluated on the Fetch Push environment in OpenAI Gym. In the experiments, however, a nonlinear function approximation is used which is not in line with the theoretical results. In addition, only simulation is used -- no real-robot experiments have been performed. To this end, the difference between the simulated and the "real" environment was the friction coefficient.

**Limitations And Societal Impact:**

The limitations of the approach are not sufficiently discussed. These should be discussed not only at the theoretical level but also with regards to real-world robotics/intelligent systems implications. That hinges however on a realistic empirical evaluation that has not been provided so far. No discussion of the societal impact was provided.

**Main Review:**

Originality: Sim to real transfer is arguably the holy grail in ML for robotics and progress along this line of research would be of great interest to the NeurIPS community. The approach presented in the paper is not necessarily novel (e.g. combining real and simulated has been tried before) but the authors present a simple method that is built upon a rigorous foundation. The rationale is very much in line with importance sampling and actually could even be extended to more environments. This could be interesting in settings where there is a simulated environment, a simplified real-world setting, and a high-fidelity real-world setting (costly samples). The approach addresses the question of different sampling speeds (throughput) in these different domains. The strongest aspect of the paper is the careful derivation of a theoretical convergence proof for the proposed method.

While I cannot guarantee correctness of the full proof, I check the individual derivation steps for soundness and was impressed by the detailed analysis in the appendix. The authors also do a good job of clearly communicating which assumptions are being made here. It would however have been interesting to discuss what ramifications these assumption have in practice for robot learning tasks (not just their theoretical background). The proof itself is non-trivial and very interesting. Some of the methodology could be used for other similar proofs going forward. Despite the theoretical rigor, the paper is easy to follow and well-written.

Unfortunately, the paper fails short when it comes to the empirical evaluation. To begin with, there is no "real" environment in the experiments. The OpenAI Gym Push environment is used twice with different friction parameters. This doesn't do the task justice and just flatly needs to be rejected as a sim-to-real experiment. Real world environments include a variety of aspects that would have put the theoretical assumptions underlying the proof to test. I would have understood such an experiment, if the paper was differently framed. But the title is literally "Sim and Real" and up to the evaluation section the reader is left to believe that the theoretical propositions will be evaluated in the real world. The fact that only the friction between the object and the table. Hence, only a small difference along a single parameter exists between the simulated and the real world. No differences in actuation, sensing, noise levels, time deltas, wear-and-tear, etc. The authors explicitly talk about reduction in sampling requirements for robotic manipulation. Given the presented empirical results, I can't see how we can draw this conclusion at the moment.

To summarize, the paper addresses an important issue and proposes an interesting theoretical framework that (if true) allow for improved performance when combining real-world and simulated data. However, the performed experiments do not allow for any sort of conclusion at the moment. Some assumptions are still not well explained. For example, the difference in transition functions between the real and simulated world needs to be bounded. What does that mean in practice? How small does this bound have to be in practice? Especially given the claims made in the beginning of the paper. I still salute the authors for the effort put into a rigorous theoretical analysis of the problem. My recommendation is to setup a set of real-world experiments that show both the strength and the limitations of the method. This could be a very strong paper, if these changes are made.

Edit after rebuttal: Thank you to the authors for addressing our concerns. However, I maintain my score. The paper in its current form is not ready for publication. The main challenge is that the topic of concern is not realistically addressed in the paper. Since the title says "Sim and Real", it is important to clearly specify and address the challenges inherent to sim2real problems in robotics. That, in my opinion, has not been achieved in this paper. An interesting theoretical formulation has been introduced, but there is no grounding of this fomalization that allows us to believe that this would solve (or even partially address) the sim2real problem. In the experiment, a mapping between two slightly differing variants of the same simulation is performed - definitely not sim to real. Many times in robotic sim2real tasks, multiple aspects (difference in geometry between sim and real, diff in parameters, diff in dynamics, invalid assumptions about rigidity of objects, etc.) compound in complex ways and render solutions, as the one proposed here, impossible. There is nothing unique in the proposed theoretical or practical implementation that suggests that a future extension of this framework would allow us to solve the true sim2real problem. My recommendation to the authors is really to ground this approach in a real-world experiment in which your assumptions can be tested compared the real world and in which the performance of your method can be reasonably evaluated.

**Time Spent Reviewing:**

12

---

> ### Author Response · Authors · 2021-08-08
> **Our main contribution is theoretical. The purpose of the experiment section is to demonstrate the behaviour of such a mixture approach.**
>
> Thank you for your feedback and for your insightful comments on our paper.
>
> ### Comment 1:
> There is no "real" environment in the experiments. The OpenAI Gym Push environment is used twice with different friction parameters..... only a small difference along a single parameter exists between the simulated and the real world. No differences in actuation, sensing, noise levels, time deltas, wear-and-tear, etc.
>
> ### Response:
> Our main contribution is theoretical: we prove the first convergence result of a mixture paradigm between sim and real samples (or any K environments samples). The purpose of the experiment section is to demonstrate the behaviour of such a mixture approach. Therefore, we chose the FetchPush environment especially for its simplicity and we changed only the friction to emphasis a main point in the paper: Even when sim and real are relatively "close" (only friction is different between the environments), learning only with sim samples may fail (as shown in the experiments section). Using only real samples solves the task, but of course, is costly. Thus, a mixture approach is required and controlling the sampling probabilities and optimization probabilities lead to higher performance - first, by solving the task. Second, solving it with less real samples.
>
> ### Comment 2:
> The difference in transition functions between the real and simulated world needs to be bounded. What does that mean in practice? How small does this bound have to be in practice?
>
> ### Response:
> In practice, it is hard to measure how the sim-real closeness is satisfied. Most real-world domains have high dimensional state and action spaces, making it impractical to measure. Although Assumption 7 is required for the theoretical part, we tried in the experiments to reflect it in the way of changing only the friction between sim and real, as we explained in the response to comment 1.
>
> ### Comment 3:
> ... The limitations should be discussed not only at the theoretical level but also with regards to real-world robotics/intelligent systems implications.
>
> ### Response:
> One major limitation of our approach is that it requires sampling from the real world during the *training* process. In real world domains it may be easier to collect in advance a fixed amount of samples from the real world and use it for training. As presented in the paper, our approach assumes a fixed rate of collecting data from the real world.

---

### Official Review · Reviewer_jQKF · 2021-07-17

**Rating:** 4
**Confidence:** 4

**Summary:**

The paper combines simulation and the real world data by maintaining a replay buffer for both. The agent selects both the environment as well as the replay buffer with respective probabilities and uses that to estimate the TD error and update the policy parameters. They test this on a simple and well known `FetchPush` task and provide a theoretical convergence analysis.

On a high level, the paper is about how to mix and balance simulation and real world data to increase the probability of the transfer of policies from simulation to the real world.

**Limitations And Societal Impact:**

The work focusses on simplified scenarios e.g. the environment `FetchPush` is a relatively easier environment and already has very little reality gap due to fewer contacts and task of nudging the object along the table (no lifting needed). The policy consequently also seems simpler requiring linear actor critic. It would make sense to try out slightly harder tasks especially where contacts are made and broken and the task requires more than just nudging the object. It would not only help testing on more challenging problem but also makes collecting more real world data harder and slow because of the complexity of the problem.

I think the choice of mixing real and sim data makes sense and has been of interest to many preceding works in this direction so overall this isn't entirely new. While they have provided a way to combine environment and replay buffer to show the trade-offs of each but I feel on a challenging tasks you'd have far less real world data and in the end need a strategy that is trained entirely on sim first and fine tuning a bit on the real world data. However, this also involves tuning the simulator so that fewer real world data is needed which in itself isn't as straightforward.

**Main Review:**

Adding real world data to the learning process whenever makes a lot of sense. In reality, there is never a zero-shot transfer so every time you train a policy in simulation you have to verify if it transfers to the real world and that means running experiments on the real world. This also means you get real world data generated through the testing and use that to improve your model or simulation parameters. Even if the policy fails to solve the task in the real world right way, the collected data is still useful to providing a snapshot of real world environment.

The paper focuses on the theoretical analysis more and therefore uses a simplified task as well as representation of the policy i.e linear actor critic. Experimental analysis on various ways to combine real and simulation data chosen based on categorical distribution shows that design choices (q_r - sampling real world environment probability and beta_r - sampling real world replay buffer probability) matter.

- For a fixed beta_r, when q_r is low, most of the data comes from sim and role of real world is only to "fine tune" the learning. When q_r increases, the data distribution is composed of two different environments which may confuse the agent.
- When beta_r is too high (with respect to qr, i.e., beta_r = 0.9) the performance degrades and that choosing beta_r > qr is preferable.
- Applying the mixing strategy after transferring the policy trained entirely in sim, to real can reduce further the required real episodes while maintaining high success rate.

**Time Spent Reviewing:**

1

---

> ### Author Response · Authors · 2021-08-08
> **We chose the FetchPush environment especially for its simplicity and we changed only the friction to emphasis a main point in the paper.**
>
> Thank you for your feedback and for your insightful comments on our paper.
>
> ### Comment 1:
> The work focuses on simplified scenarios e.g. the environment FetchPush is a relatively easier environment and already has very little reality gap due to fewer contacts and task of nudging the object along the table (no lifting needed).... It would make sense to try out slightly harder tasks....
>
> ### Response:
> We chose the FetchPush environment especially for its simplicity and we changed only the friction to emphasis a main point in the paper. Even when sim and real are relatively "close" (only friction is different between the environments), learning only with sim samples may fail (as shown in the experiments section). Using only real samples solves the task, but of course, is costly. Thus, a mixture approach is required and controlling the sampling probabilities and optimization probabilities lead to higher performance - first, by solving the task. Second, solving it with less real samples.
>
> ### Comment 2:
> I think the choice of mixing real and sim data makes sense and has been of interest to many preceding works in this direction so overall this isn't entirely new. While they have provided a way to combine environment and replay buffer to show the trade-offs of each but I feel on a challenging tasks you'd have far less real world data and in the end need a strategy that is trained entirely on sim first and fine tuning a bit on the real world data. However, this also involves tuning the simulator so that fewer real world data is needed which in itself isn't as straightforward.
>
> ### Response:
> You are right. Mixing sim and real samples is not a new idea. However, other methods that involved both sim and real do not control the mixing rate. Most of them use sim first and then fine tune on the real environment. As you mentioned, many times this approach requires tuning the simulator. Our method does not require that. Our claim is that if sim and real are "close" (as discussed in the previous comment), controlling the mixing between sim and real (as described in the paper) converges to a fixed point.

---

### Official Review · Reviewer_7FTV · 2021-07-17

**Rating:** 6
**Confidence:** 3

**Summary:**

This is a paper on sim-to-real transfer, proposing a new method for combining data from both the sim and real domains in order to obtain a better tradeoff between sample efficiency and quality of learning. This is a natural and good idea, although it appears novel in this proposed form.

The technical contribution is to devise a scheme to store data in multiple replay buffers, corresponding to the different sim and real sources, then using this data in a mixed linear actor critic scheme by suitably sampling the source and data points online. This paper attempts to present an analysis of the mixing scheme to argue convergence despite the difference in environments, as long as the sim and real domains are not too different in a certain sense. Finally, there is an empirical evaluation involving multiple simulations of a simple robotic pushing task.



**Ethical Concerns:**

-

**Limitations And Societal Impact:**

As my comments make clear, the paper could be strengthened in terms of consideration of limitations and the plausibility of assumptions. I say this while acknowledging that the core analysis is rigorous and appears sound - but the overall aim is to address what is ultimately a practical issue about the real world. A section on these issues would be good to include.

This is a methodological contribution to learning algorithms, which does not immediately impact societal issues.

**Main Review:**

This is a timely problem, which the authors of the paper handle competently. The key technical steps have to do with controlling the collection and sampling processes, and the consequent effect on approximations within the actor-critic scheme. The fact that the agent is actively switching between different environments and using different replay buffers to update the policy calls into question the convergence of such an algorithm. This is mitigated by noting that the environments are not too dissimilar and by tuning the ratios of sampling rates.

I have a few different points to query, starting with the experimental framing. The version of this work that utilises the "real" environment has only one sim and one real environment, where the real environment is really a more noisy simulation of the sim environment. With this, the initial experiments show that a certain level of sampling (q_r and \beta_r) are required for good performance. However, this seems delicate. I note the phenomenon wherein increasing q_r actually makes the learning deteriorate (unless it is a very low level indeed, at 0.1). The explanation given is that the mixed nature of the samples confuses the agent, but this really is the test and raises the question of whether a more informed schedule is needed for the mixing. This is acknowledged in the mixing section, where the conclusion is that introducing mixing after transferring to real is beneficial. I would have liked to have seen a lot more detail on these options and whether the proper procedure requires optimization of #5 by changing the interleaving of mixing, sim and real.

On a similar note, it seems like one issue here is that the sim environment might not be capturing the real diversity of phenomena in the real world. So, it would be good to understand the behaviour of such a scheme when there are multiple sim environments (i.e., K >> 2), representing significant diversity. This would address two types of questions - how do the relative numbers for q_r and \beta_r impact performance in this more realistic setting, and how well does the procedure scale when the real environment is substantially different from sim but different sim environments capture different parts of this difference well.

The next point that concerns me is the assumption about the nearness of sim and real. This paper rests on the strong assumption that the only difference between the two is a benign form of noise, which is why assumption 7 is reasonable. However, in real applications - especially in robotics - there are other forms of differences. For instance, there is state-action dependent noise such as objects behaving significantly differently in low or high parameter regimes, or collisions and deformations being wrongly simulated, and so on. In these cases, there will not be a single \epsilon_s2r that makes sense (or it would be too large to be useful). If this assumption were to be relaxed to be more realistic, I wonder how much this changes the claims of Theorem 4. We are not just asking what a larger \epsilon means, which is clear. Instead, we are asking if assumption 7 were to be redefined in terms of different states being at different levels of fidelity, and we consider this in the convergence analysis, what is the effect on the training dynamics.

The fact that the sim2real experiment is really a sim2sim' makes it such that these issues have not arisen to a large extent in the experiments. The paper would be stronger if the experiments really did consider a *real* real environment (with realistic deviations between sim and real), so that one could get an empirical understanding of such issues.

Lastly, on a stylistic note, I found it a bit confusing that distributions and random variables are called by the same symbols in different places, e.g., \beta in the algorithm description. Could this be revisited to clarify?



**Time Spent Reviewing:**

2

---

> ### Author Response · Authors · 2021-08-08
> **Our contribution is primarily theoretical. The purpose of the experiment section is to demonstrate the behaviour of such a mixture approach.**
>
> Thank you for your feedback and for your insightful comments on our paper.
>
> ### Comment 1:
> ...I would have liked to have seen a lot more detail on these options and whether the proper procedure requires optimization of #5 by changing the interleaving of mixing, sim and real.
>
> ### Response:
> We conducted the experiments following the theoretical analysis where the probabilities for sim and real were fixed during the learning process. Learning a policy for changing these probabilities during training is very interesting, but it was out of the scope of the paper. An interesting future work can focus on learning methods, for example exploit the gradient or Bellman error  for both sim and real and change the probabilities to minimize them.
>
> ### Comment 2:
> The sim environment might not be capturing the real diversity of phenomena in the real world. So, it would be good to understand the behaviour of such a scheme when there are multiple sim environments (i.e., K >> 2), representing significant diversity. This would address two types of questions - how do the relative numbers for q_r and \beta_r impact performance in this more realistic setting, and how well does the procedure scale when the real environment is substantially different from sim but different sim environments capture different parts of this difference well.
>
> ### Response:
> We agree that the situation of one or several sim environments that are substantially different from the real environment is a realistic setting. However, it violates a main assumption in our work - that sim and real are relatively close. And even for multiple sim environments, they all need to be "close" enough to the real environment.
>
> ### Comment 3:
> ...The assumption about the nearness of sim and real. This paper rests on the strong assumption that the only difference between the two is a benign form of noise, which is why assumption 7 is reasonable. However, in real applications - especially in robotics - there are other forms of differences.....  If this assumption were to be relaxed to be more realistic, I wonder how much this changes the claims of Theorem 4. We are not just asking what a larger \epsilon means, which is clear. Instead, we are asking if assumption 7 were to be redefined in terms of different states being at different levels of fidelity, and we consider this in the convergence analysis, what is the effect on the training dynamics.
>
> ### Response:
> Your idea for splitting the "closeness" between sim in real to different levels of fidelity is very interesting. We think that it might be possible to create a state-dependent metric such that if many transitions to states have low difference between sim and real, they can “cancel out” few transitions with high difference. This is an excellent direction to future work.
>
> ### Comment 4:
> The fact that the sim2real experiment is really a sim2sim' makes it such that these issues have not arisen to a large extent in the experiments. The paper would be stronger if the experiments really did consider a real real environment (with realistic deviations between sim and real), so that one could get an empirical understanding of such issues.
>
> ### Response:
> Our contribution is primarily theoretical, where we, for the first time, prove the convergence of a mixture approach between sim and real samples (or any K environments samples). The purpose of the experiment section is to demonstrate the behaviour of such a mixture approach. Therefore, we chose the FetchPush environment especially for its simplicity and we changed only the friction to emphasis a main point in the paper - Even when sim and real are relatively "close" (only friction is different between the environments), learning only with sim samples may fail (as shown in the experiments section). Using only real samples solves the task, but of course, is costly. Thus, a mixture approach is required and controlling the sampling probabilities and optimization probabilities lead to higher performance - first, by solving the task. Second, solving it with less real samples.
>
> ### Comment 5:
> I found it a bit confusing that distributions and random variables are called by the same symbols in different places, e.g., \beta in the algorithm description. Could this be revisited to clarify?
>
> ### Response:
> Thank you for the note. We will go over the notations and symbols and clarify where needed.

---

### Official Review · Reviewer_r88y · 2021-07-19

**Rating:** 7
**Confidence:** 3

**Summary:**

The paper presents a reply-buffer based algorithm to learn simultaneously from multiple environments. The target application is learning in a mixture of simulated and real environments but the theoretical analysis is conducted in the context of multiple environments. The proposed approach is sim-real-mix which maintains a reply buffer for each of the available environments; actor and critic updates are computed with a TD error which results from a random sampling from the available reply buffers according to a static categorial distribution.

The proposed sim-real-mix algorithm (theoretical analysis limited to linear actor-critic) is proved to converge to fixed points for both the actor and the critic. Additionally, bounds on the fixed points are given in case the case of two environments (i.e. sim and real) for which a distance between the transition probabilities is upper bounded.

Additionally, the authors propose some experimental evaluations of their sim-real-mix algorithm, testing it on two slightly different instances of the same simulated environments. The idea is to simulate the situation which occurs when learning from two environments (e.g. one sim environment and one real environment) with the aim of solving a task in one environment (i.e. the real environment) while taking advantage of data produced by both environments, assumed to share some similarities. The experimental evaluation aims at understanding how the proposed sim-real-mix algorithm behaves when varying the categorical distributions used for sampling (Figure 2 and 3). Additionally, authors compare the proposed strategy with a standard sim2real fine tuning (i.e. learn in sim and fine-tuning in real) and with a modified sim2real (i.e. learn and sim and fine-tune with the proposed sim-real-mix strategy). Interestingly it is observed that fine-tuning with sim-real-mix data is better than fine-tuning with real data only.

**Ethical Concerns:**

No ethical concerns.

**Limitations And Societal Impact:**

No considerations needed.

**Main Review:**

The submitted paper is original, sound, clearly written and significant to the field. The relevance to field is mainly theoretical but I don't exclude that in the future this theoretical analysis can be useful for practical considerations. The major limitation of the submitted paper is that the theoretical results hold only for linear function approximation even though interesting applications would require the extension to non-linear neural networks. Another important limitation is that the main results on sim/real convergence (Section 5.2) rely on the assumption of being able to measure the sim-real closeness (as defined in Assumption 7) which however can be very impractical. The sim-real closeness is given as |P_sim(s'|s, a) - P_real(s'|s, a)| < epsilon which is impractical since it's in general difficult to measure the state of a real system. On real systems the state is often a function of on non-directly-observable quantities and therefore the sim-real closeness would be more practically characterized in terms of observations only, rather than relying on states.

**Time Spent Reviewing:**

3.5

---

> ### Author Response · Authors · 2021-08-08
> **We demonstrated the "closeness" between sim and real by changing only the friction between the environments.**
>
> Thank you for your feedback and for the insightful comments on our paper.
>
> ### Comment 1:
> The theoretical results hold only for linear function approximation even though interesting applications would require the extension to non-linear neural networks.
>
> ### Response:
> In reinforcement learning many theoretical convergence analysis tools are based on linear actor-critic algorithms. In recent years, a step forward was made and tools for analyzing non linear (i.e., neural networks based) RL algorithms were developed (for example, [1]). In our analysis we “keep it simple" regarding the linear approximation aspect of the algorithm, and focus on novel analysis methods regarding our contributions in the paper: using K environments, sampling from a replay buffer, using different sampling probabilities for collecting data and optimizing the agent.
>
> [1] A Theoretical Analysis of Deep Q-Learning, Fan, Jianqing and Wang, Zhaoran and Xie, Yuchen and Yang, Zhuoran, 2020.
>
> ### Comment 2:
> The main results on sim/real convergence (Section 5.2) rely on the assumption of being able to measure the sim-real closeness (as defined in Assumption 7) which however can be very impractical.
>
> ### Response:
> We agree that the measure of sim-real closeness is impractical in real world domains, where state and action spaces are large and where we do not know the exact transition probabilities. However, this measure is required for theoretical purposes and we tried to demonstrate its meaning in a simple experiment where we changed only the friction between sim and real. Changing the friction slightly affects the transition probabilities between states. Hence, sim and real are "close" in this sense.

---

### Decision · Program_Chairs · 2021-09-27

**Decision:**

Accept (Poster)

**Comment:**

The subject of this paper is sim2real transfer, and it tackles this with two different parts. Firstly, it proposes an algorithm for sampling interactions from two different replay buffers, a simulated and a real one, the objective being to sample far less samples from the real environment, where interactions are costly. The algorithm is evaluated in an experimental part. Secondly, the paper contains a theoretical part with a congergence analysis of the provided algorithm.

This paper always was on the fence and was discussed by 4 reviewers, who essentially agreed on the strengths and weaknesses of the paper, but who disagreed on how to weight them and on the final decision.

All reviewers agreed that the algorithm itself was not sufficiently novel and did exist in some form in the literature. They also agreed on the weaknesses of the experimental part, in particular the simplicity of the chosen tasks and environments.

It then boiled down to whether the theoretical insights (convergence properties) are of interest to the community and whether they justify accepting. Here, reviewers were split 50/50.

The AC agreed that this paper is on the fence but judges that the theoretical derivations are of value and merit publishing in NeurIPS.
This paper was discussed between the AC and SAC.